# Quantifying the Impact of Street Greening during Full-Leaf Seasons on Emotional Perception: Guidelines for Resident Well-Being

**Nayi Hao [1], Xinzhou Li [2], Danping Han [3],* and Wenbin Nie [1],***

[1] College of Landscape Architecture, Zhejiang A&F University, Hangzhou 311300, China; nyhao827@gmail.com
[2] Department of Landscape Architecture, The University of Sheffield, Sheffield S10 2TN, UK; jingdelixinzhou@gmail.com
[3] College of Urban Construction, Zhejiang Shuren University, Hangzhou 311300, China
* Correspondence: 601484@zjsru.edu.cn (D.H.); niewenbin@zafu.edu.cn (W.N.)

**Abstract:** Quantifying the emotional impact of street greening during the full-leaf seasons in spring, summer, and fall is important for well-being-focused urban construction. Current emotional perception models usually focus on the influence of objects identified through semantic segmentation of street view images and lack explanation. Therefore, interpretability models that quantify street greening's emotional effects are needed. This study aims to measure and explain the influence of street greening on emotions to help urban planners make decisions. This would improve the living environment, foster positive emotions, and help residents recover from negative emotions. In Hangzhou, China, we used the Baidu Map API to obtain street view images when plants were in the full-leaf state. Semantic segmentation was used to separate plant parts from street view images, enabling the calculation of the Green View Index, Plant Level Diversity, Plant Color Richness, and Tree–Sky View Factor. We created a dataset specifically designed for the purpose of emotional perception, including four distinct categories: pleasure, relaxation, boredom, and anxiety. This dataset was generated through a combination of machine learning algorithms and human evaluation. Scores range from 1 to 5, with higher values indicating stronger emotions and lower values indicating less intense ones. The random forest model and Shapley Additive Explanation (SHAP) algorithm were employed to identify the key indicators that affect emotions. Emotions were most affected by the Plant Level Diversity and Green View Index. These indicators and emotions have an intricate non-linear relationship. Specifically, a higher Green View Index (often indicating the presence of 20–35 fully grown trees within a 200 m range in street view images) and a greater Plant Level Diversity significantly promoted positive emotional responses. Our study provided local planning departments with support for well-being-focused urban planning and renewal decisions. Based on our research, we recommend the following actions: (1) increase the amount of visible green in areas with a low Green View Index; (2) plant seasonal and flowering plants like camellia, ginkgo, and goldenrain trees to enhance the diversity and colors; (3) trim plants in areas with low safety perception to improve visibility; (4) introduce evergreen plants like cinnamomum camphor, osmanthus, and pine.

**Keywords:** street view images; semantic segmentation; human-machine adversarial scoring; emotional perception driving factors; Plant Level Diversity; Green View Index; interpretable machine learning; Shapley value

## 1. Introduction

It is crucial to systematically measure and clarify the impact of street greening during the full-leaf seasons in spring, summer, and fall on individual emotions in order to create a more favorable and healthier living environment. Exposure to urban green spaces can exert a favorable influence on individuals' emotional well-being by providing benefits such as mitigating stress and enhancing happiness [1–3]. Exploring the influence mechanism

of "human-scale" urban green spaces on human emotions is a crucial aspect of research when examining the relationship between humans and the urban built environment [4]. Streets, which are the most widely distributed public spaces inside urban areas, exhibit a close connection with the daily lives of residents [5]. The street-side distribution of trees, shrubs, and ground cover plants can offer residents easily accessible green spaces, which is a significant factor that influences human emotional perception [6]. In contemporary high-density urban environments [7], there is a discernible trend of individuals toward an increasingly accelerated pace of life. The convenience and accessibility of visiting street green space surpass those of other green places, such as parks. Therefore, it is critical to evaluate the quality of street green space from the perspective of human vision as well as the relationship between accessing street green space and human emotion perception. Furthermore, this evaluation will improve residents' well-being and advance urban development that prioritizes human needs [8,9].

Studies on urban green space typically employ techniques such as Global Positioning System (GPS), Geographic Information System (GIS), field investigation, street view images, and other methods to gather data on vegetation [10–13]. However, it is difficult for a GIS analysis based on remote sensing images to measure green exposure at the street scale, while the utilization of GPS tracking mobile devices and field surveys is constrained by limited sample sizes [14]. With the emergence of geographic big data in recent years, street view images have been widely used in urban research [15]. Street view images, which are photos shot along urban streets from the human perspective, contain a wealth of visual information about the urban road environment and can effectively reflect scenery seen in people's daily lives as well as the physical characteristics of the city [16]. In contrast to alternative spatial and temporal data sources such as conventional remote sensing satellite images, street view images more closely align with the perception of the surrounding street environment by human vision [14]. Furthermore, these images can be integrated with deep learning and other artificial intelligence technologies, as well as semantic segmentation and other computer vision technologies, to expand the potential applications of urban quantitative research. The machine learning algorithm, which is advancing quickly, can precisely and effectively detect and extract green elements in street view photos, thereby enhancing the precision of measuring greening. Simultaneously, utilizing extensive street view data from mapping sources like Google Maps enables the rapid assessment of large-scale street greening. Scholars have evaluated people's subjective perceptions of urban built environments from different perspectives based on street view images and computer technology [17–19]. These studies have demonstrated the efficacy of investigating the impact of physical features within urban environments on human perception and psychological well-being using street view images and computer technologies. Among these, research on street greening involves measuring street greening accessibility [9], evaluating the connection between safety perception and green space visibility [20,21], and examining how street trees lower the risk of depression [22]. Although there has been increasing research on the impact of street greening on emotional perception and the positive effects of street greening on people's emotions, such as increasing happiness and reducing stress, which have been confirmed, there are still two issues in the relevant research.

How street greening drives human emotions remains an open question because of limited empirical evidence. To study the driving mechanism of street greening on emotions, it is essential to perform an initial quantitative assessment of street greening. The greening evaluation index, as a factor that quantifies the level of street greening, plays a mediating role in our research, connecting emotional perception and street greening. At present, the main methods for assessing urban greening include the normalized difference vegetation index (NDVI), fractional vegetation cover (FVC), Green View Index (GVI), and others [23–26]. NDVI and FVC, as indicators from a top–down perspective that use remote sensing satellite images, cannot effectively represent the level of greenery seen from a human perspective at the street scale, while GVI can compensate for the shortcomings of NDVI and FVC by using street view images as data. Furthermore, as proposed by Yang et al., the evaluation of urban

greening should incorporate various factors and vertical dimension greening indicators to assess the extent of visible street greening [27]. Hence, considering the three dimensions of ecology, society, and landscape [28], we comprehensively considered whether these indicators are well represented in street view images, how they can be quantitatively measured, whether they are clearly perceived by people, and other factors. Ultimately, we selected various indicators, including the Plant Color Richness, Plant Level Diversity, Green View Index, and line of sight, as the means to assess the effectiveness of the street greening system.

In addition, there is a lack of adequate research on how to quantify and explain the impact of street greening on emotional perception. Currently, numerous studies employ linear models to quantitatively examine the influence of the environment on perception [29–32]. However, the connection between human emotions and the external environment is often intricate [33–35]. Existing research often presumes that there is a linear link between human emotion perception and the environment. Nevertheless, we contend that this assumption may be insufficient to fully elucidate the profound influence of a green environment on emotions [36]. With the rapid development of the field of machine learning, interpretability methods such as Shapley Additive Explanation (SHAP) have been introduced into urban research [37–41]. Given its superior interpretation capability and consistency with human intuition, we opted to employ the SHAP interpretation method to investigate the influence of street greening indicators on emotion perception, both in terms of intensity and form. By doing so, we aim to gain insights into the underlying mechanisms through which street greening affects emotion perception.

To accomplish the abovementioned goals, our study focused on Hangzhou, China, as the research area and constructed an interpretable model based on computer vision technology and deep learning technology to assess the influence of street greening during the full-leaf seasons in spring, summer, and fall on emotional perception. We acquired street greening data through the process of semantic segmentation of crawled street view images. Using these data, we quantitatively assessed street greening by computing the values of several greening indicators, which served as explanatory variables for the perception model. Simultaneously, a sentiment perception dataset was created by merging the outcomes of manual scoring and machine learning, including four distinct categories: pleasure, relaxation, boredom, and anxiety. The scoring system ranges from 1 to 5, with higher values indicating more intense emotional experiences and lower values suggesting less intense sensations. This sentiment perception dataset was subsequently employed as the dependent variable in the perception model. The study successfully identified critical indications of greening that have a significant impact on emotions using the random forest algorithm and SHAP algorithm. Additionally, the research explores the non-linear correlation between various indicators of greening and emotional perception. The research we conducted collected data from four different levels, specifically the Green View Index, Plant Level Diversity, Plant Color Richness, and Tree–Sky View. These data are intended to assist local planning departments in making decisions regarding urban planning and renewal that prioritize the well-being of residents. Additionally, they have the potential to create a more pleasant and healthy environment, motivating inhabitants to cultivate positive emotions and recover from negative ones.

## 2. Methods and Materials

### 2.1. Study Area

Hangzhou, as a highly developed location in China in terms of its socioeconomic status, is also one of the earliest cities to be selected as a "National Garden City of China", which represents an exemplary level of urban greening in China [42]. Our research selected several major districts in Hangzhou, Zhejiang Province, China, as research areas, including the Xihu, Gongshu, and Binjiang Districts. These regions in Hangzhou are the primary urban centers and have a long history of development. The streets and other infrastructure in these areas are well established, and the street greening is highly representative. Their

green environment is rather stable, but there is a need for renovation and enhancement. Hence, we selected these areas as the focus of our research (Figure 1).

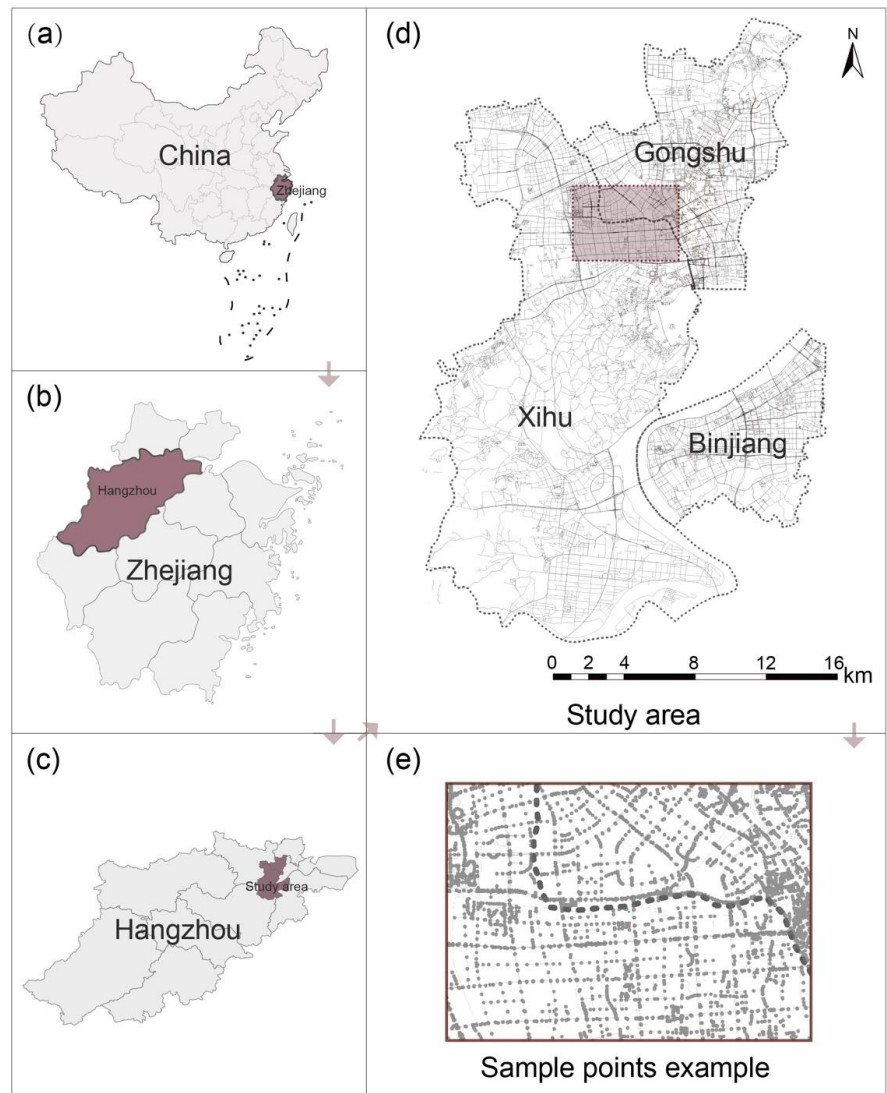

**Figure 1.** Overview of the research area: (**a**) location of Zhejiang Province in China, (**b**) location of Hangzhou City in Zhejiang Province, (**c**) location of the Xihu, Gongshu, and Binjiang Districts in Hangzhou, (**d**) road network of the Xihu, Gongshu, and Binjiang Districts, (**e**) example of sample points.

*2.2. Research Framework*

A schematic representation of the study methodology is depicted in Figure 2. This methodology uses computer vision and deep learning methodologies to evaluate and analyze the specific effect of street greening on human emotional perception. The study involves four main components: (1) the acquisition of street view images within the research area; (2) the application of DeepLabv3+ for segmenting these images to determine the proportion of different objects; special attention was given to street plants, and quantitative measurements of plant landscape indicators such as plant-level richness were calculated; (3) evaluation of the perception score of street plants through panoramic street images, followed by the generation of an emotional perception dataset using a human-machine adversarial approach; and (4) development of a random forest regression model through combining the perceptual score data and the quantitative results of plant landscape indicators. Subsequently, the SHAP method was employed to elucidate the mechanism by which plant landscape indicators influence human emotion perception.

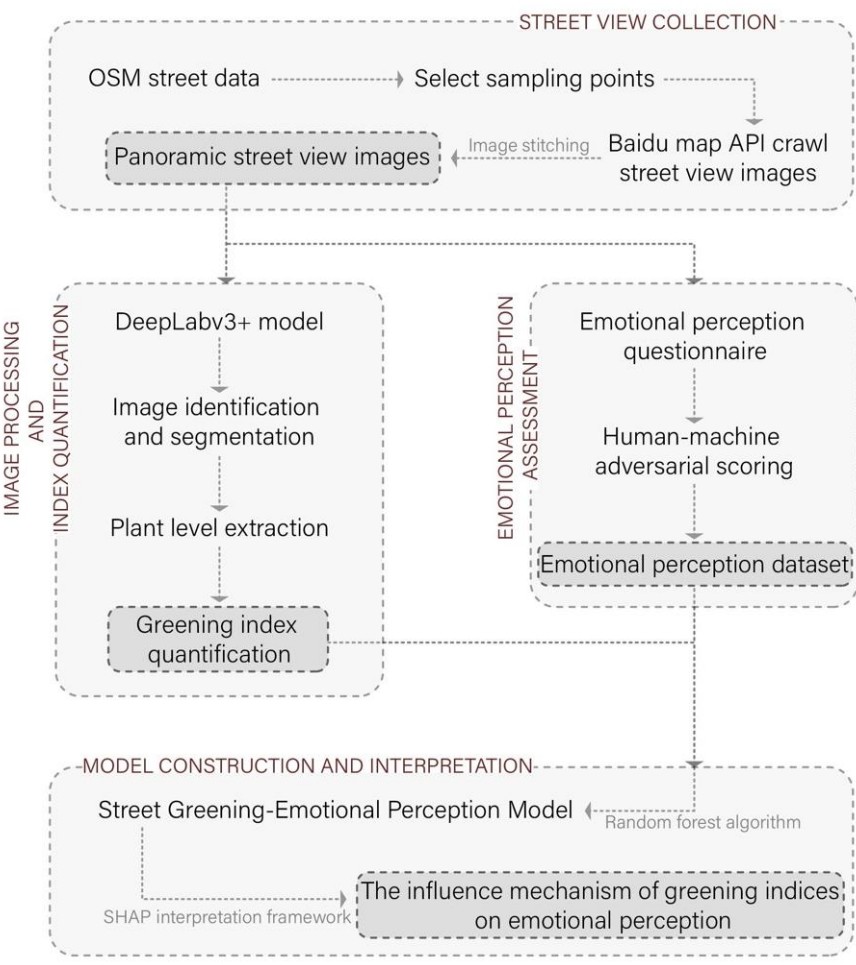

**Figure 2.** Research flow.

*2.3. Data and Data Preprocessing*

2.3.1. Street View Image Acquisition

First, we used Open Street Map to collect the area's street network, and then we employed ArcGIS to create a series of observation points at regular intervals along the chosen route. Sample points were chosen at intervals of 200 m for streets exceeding a length of 200 m, whereas streets with a length less than 200 m were subject to sample points at intervals of 100 m. A total of 46,453 samples were taken. Subsequently, the street view image was systematically retrieved at every designated sample location using the Baidu Map API. The street view images captured in our study were exclusively taken from September 2017 to June 2020. To mitigate the effects of seasonal variations on the plant scenery, we substituted the winter photos of Hangzhou (from December to February of the subsequent year) using the Timestamp feature provided by Baidu Map. To comprehensively illustrate the presence of vegetation on the streets within the designated study area, we procured four street view images from each sampling point by conducting a visual survey in four cardinal directions (0°, 90°, 180°, 270°) from a human-eye-level perspective (vertical angle was set to 20°). The dimensions of each image are 960 by 720 pixels, with a pixel density of 72 pixels per inch. Subsequently, these four images were merged to create a panoramic picture encompassing the surroundings of the respective sampling point (Figure 3). Panoramic images offer comprehensive visual data and more accurately replicate individuals' perceptual experiences in the actual surroundings. Furthermore, the utilization of panoramic images might mitigate the possible influence of landscape alterations from different directions on the results of the experiment, hence guaranteeing the reliability and replicability of our research.

**Figure 3.** A panoramic image is created by merging street view photographs captured from four different viewpoints.

### 2.3.2. Preprocessing of Street View Images

The DeepLabv3+ neural network technique was employed in this study to perform semantic segmentation and recognition of street view images [43]. DeepLabv3+ is an optimized algorithm based on the principles of atrous convolution and the conditional random field. The proposed method effectively analyses the global structural information of picture pixels as well as the local spectral information of pixels, enabling reliable identification of varied shapes and even partially deformed photographs. Past research has demonstrated impressive outcomes in the semantic segmentation of street view images using DeepLabV3+, and the approach has also exhibited strong practical performance [42,44–46]. In terms of the dataset, we selected the ADE20K dataset [47], which contains 150 categories of objects. The ADE20K dataset possesses the attributes of extensive picture coverage, superior semantic segmentation and annotation resolution, as well as a diverse range of scene and object categories. In contrast to alternative datasets that exclusively encompass trees and vegetation within the plant category, the ADE20K dataset additionally encompasses shrubs and flowers, which is essential to our research. The ADE20K dataset has been extensively utilized and has yielded notable outcomes in the realm of urban research in China [48–51]. This suggests that the semantic segmentation annotation of ADE20K effectively captures the intricate and varied characteristics of the urban landscape in China. In this study, we focused on the sky, grass, flowers, trees, and other elements. The collected street view images were processed using DeepLabv3+ to classify them into distinct color groups using the decoder. The types of plants, such as lawns and trees, in the images could be clearly extracted.

### 2.3.3. Sample Population for the Emotional Perception Survey

We recruited 82 persons residing in the Xihu, Gongshu, and Binjiang Districts to participate in our study. Volunteers were required to fulfil the condition of possessing a residency in Hangzhou for a minimum duration of 6 months. The demographic information of the participants enlisted in the study is presented in Table 1. The mean age of the volunteers enrolled in this study was 32.50 years. Their ages span from 18 to 55. From a gender standpoint, the male volunteers constitute 53.66% of the total, which is slightly higher than the percentage of female volunteers (46.34%). In relation to education level, the percentages of individuals with primary school education or below, middle school and high school education, and college school or higher education are 25.61%, 40.24%, and 34.15%, respectively. The Han Chinese population accounts for 98.78% of the total. Volunteers were instructed to evaluate a set of 500–1000 street view images that were chosen at random using the four dimensions indicated above. Based on the 5-level Likert scale [52], we established a scoring range of 1–5 points for each dimension. A score of 1 indicates the lowest degree, while a score of 5 indicates the maximum degree (Figure 4).

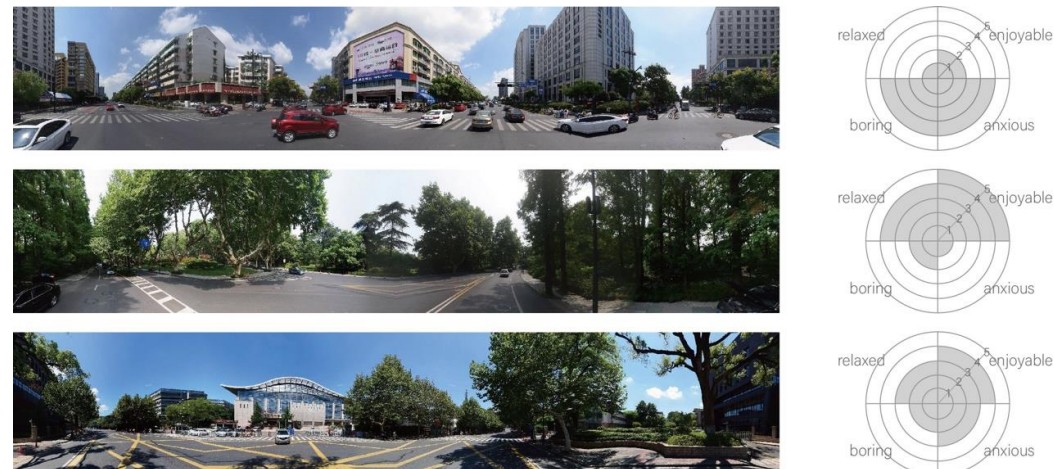

**Figure 4.** The street view images were rated based on four emotional perception perspectives.

**Table 1.** Descriptive data regarding the participants.

| Variables | Proportion/Mean (SD) |
| --- | --- |
| Age | 32.50 |
| Age Distribution (%) | |
| 18–25 | 28.05 |
| 26–35 | 32.93 |
| 36–45 | 18.29 |
| 46–55 | 20.73 |
| Gender (%) | |
| Male | 53.66 |
| Female | 46.34 |
| Education (%) | |
| Primary school or below | 25.61 |
| Middle school and high school | 40.24 |
| College or high school | 34.15 |
| Race (%) | |
| Chinese | 98.78 |
| Others | 1.22 |

### 2.3.4. Perception Evaluations from Inhabitants

The perception of the external environment is a subjective view that can be impacted by the local culture and socioeconomic background [53]. While several perception datasets have been established globally, such as MIT's Place Pulse project, it is crucial to acknowledge the disparities that exist throughout different cities. To conduct a comprehensive assessment of the perception of a certain region, it is imperative to gather perception evaluations from inhabitants who possess a deep understanding of the local background [17].

### 2.3.5. The Collection of Emotional Perception Datasets

Relevant studies on human emotional perception in the field of environmental psychology have demonstrated that the impact of the external environment on human emotions may be seen through several indicators, such as emotional intensity, emotional arousal [54], attentional processes [55], and emotional recovery [56]. In conjunction with the appropriate emotional measurement scale [57,58], we chose four specific emotions—pleasure, relaxation, boredom, and anxiety—as indicators for assessing emotional perception. Pleasure and relaxation serve as positive indicators for evaluating emotions, while boredom and anxiety serve as negative indicators. Additionally, relaxation and boredom also function as indicators for evaluating the dimensions of attention restoration and arousal. We focused

on the emotion of pleasure to understand whether greenery can create a positively delightful urban experience. The emphasis on the relaxed emotion highlighted the potential role of greenery in creating a comfortable urban environment. Simultaneously, the study of boredom underscored the impact of the environment on individual interest, and the examination of anxiety emphasized the potential effectiveness of greenery in alleviating urban stress.

Our research used a human-machine adversarial emotional perception scoring method based on deep learning technology [17], which helped us evaluate emotional perception scores relatively accurately and efficiently. Simultaneously, our scoring system incorporated the random forest method to implement an iterative feedback mechanism. As the quantity of images evaluated by participants increased, our rating system learned the rating results of previous participants. Consequently, it generated a suggested score for each image and dynamically adjusted this suggested score based on the disparity between the actual rating and the suggested score. Additionally, users were given the ability to modify the suggested score for each image. Finally, the emotional perception dataset of the study area was obtained, which could be used to establish the subsequent perception model.

2.3.6. Quantification of Greening Indexes
Plant Color Richness

Color can bring a very intuitive visual sensation, and the richness of plant color, which serves as a readily perceived indicator of plant landscape, can be measured using visual entropy. The method proposed by Han to calculate the color richness index of plants by multiplying a color factor M by the probability Pi of the formula for calculating visual entropy was proven to be feasible in the research [59]. Following the semantic segmentation of the street view images, we extracted the plant parts. Afterwards, a Python script was employed to convert the image to grayscale. The Plant Color Richness index was then computed using the provided formula. Formulas (1)–(6) illustrate the procedure for calculating the Plant Color Richness, denoted as *C* [59].

$$rg = R - G \tag{1}$$

$$\mathrm{yb} = \frac{1}{2}(R + G) - B \tag{2}$$

$$\sigma_{rgyb} = \sqrt{\sigma_{rg}^2 + \sigma_{yb}^2} \tag{3}$$

$$\mu_{rgyb} = \sqrt{\mu_{rg}^2 + \mu_{yb}^2} \tag{4}$$

$$M = \sigma_{rgyb} + 0.3\mu_{rgyb} \tag{5}$$

$$C = \sum_{i=1}^{n} M \cdot P_i \lg P_i \tag{6}$$

In the RGB color mode, *R*, *G*, and *B* represent the color values of the red, green, and blue channels, respectively. *rg* quantifies the difference between the red and green color channels, and Formula (1) is capable of capturing the alterations in red and green colors inside the image. *yb* assesses the difference between the luminosity and the blue channel, and Formula (2) is employed to measure the brightness and blue deviation of the image. $\sigma_{rg}$ and $\sigma_{yb}$ denote the standard deviations of the parameters *rg* and *yb*, respectively, while $\mu_{rg}$ and $\mu_{yb}$ represent the mean values of these parameters. Formula (3) measures the variability of the color distribution, and Formula (4) provides the central location of the color distribution. *M* is a color factor that is introduced, and $P_i$ signifies the probability of occurrence of the *i* category. This process considers the complexity of the various color distributions in the image, providing the overall color richness of the image.

Plant Landscape-Level Diversity

Diversity at the plant landscape level is the cross embodiment of the ornamental and ecological nature of the plant landscape, which can be expressed by measuring the balance and richness of plants at different levels in vertical space. After the semantic segmentation of street view images, the plant landscape can be relatively accurately divided into grass, flowers, shrubs, and trees, which provides great convenience for quantifying the diversity of plant levels. The quantification of plant landscape-level diversity can be achieved through the utilization of various indexes, such as the richness index, Shannon entropy index, and Simpson index [60], which are based on generalized entropy [61]. The richness index indicates the number of plant community levels, the Shannon entropy index implies the overall diversity of the plant landscape, and the Simpson index shows the degree of equilibrium between plant communities. Their formulas, Formulas (7)–(9), are as follows:

$$\text{Level Diversity (richness)} = N \tag{7}$$

$$\text{Level Diversity (entropy)} = -\sum_{i=1}^{N} P_i \log_2(P_i) \tag{8}$$

$$\text{Level Diversity (simpson)} = 1 - \sum_{i=1}^{N} \left(\frac{P_i}{P}\right)^2 \tag{9}$$

The variable N denotes the quantity of plant levels, including grass, flowers, plants, and trees, generated by the process of street view image segmentation, and the possible values are {0, 1, 2, 3, 4}. The larger the $N$ is, the more diverse the types of plant communities and the more levels in vertical space. The variable "$i$" represents the number of levels. In this study, we establish the following definitions: $i = 1$ denotes the level of trees, $i = 2$ indicates the level of plants, $i = 3$ represents the level of grasses, and $i = 4$ symbolizes the level of flowers. $P_i$ indicates the proportion of level $i$ in the overall number of categories following the segmentation process. Additionally, $P$ represents the cumulative proportion of all levels, calculated as the sum of $P_1$, $P_2$, $P_3$, and $P_4$. The computed results of Plant Level Diversity are depicted in Figure 5.

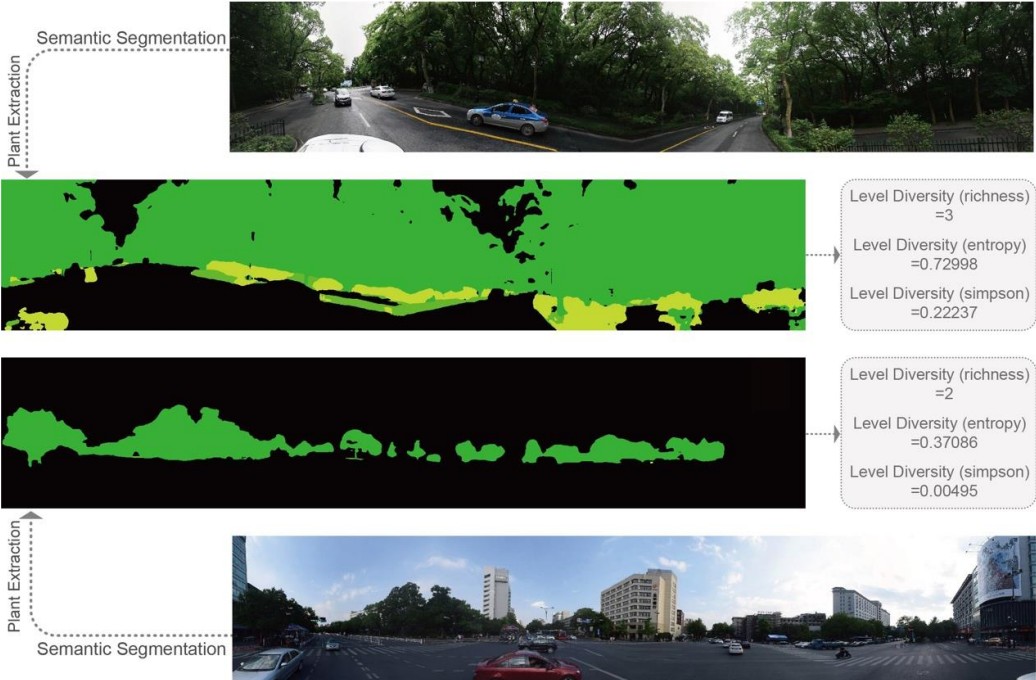

**Figure 5.** Calculation process of Plant Level Diversity.

Tree–Sky View Factor

In our study, the ratio of plants to the sky in vertical space is defined as the Tree–Sky View Factor. Prior research frequently employed fisheye images for the purpose of quantifying the sky view factor (SVF), which denotes the extent to which an individual can see the sky in a standing position. The generation of fisheye photos depicting this position could be achieved through the conversion of street view photographs into isometric azimuth projections, a method that has been empirically demonstrated to be successful [62]. However, fisheye images represent what people see when they look up, whereas our study focuses more on what people see from their natural perspective when they walk along the street. A study conducted by Bargh et al. revealed that while strolling, neighborhood residents frequently follow the street layout and rarely gaze upwards [63]. Hence, we opted to utilize segmented panoramic photos of the street view captured from a natural viewpoint to quantify the T-SVF. The calculation formula, Formula (10), is as follows:

$$TVF = \frac{Area_t}{Area_s} \tag{10}$$

The variable "$Area_t$" represents the aggregate number of pixels occupied by the tree elements in the given picture, and the variable "$Area_s$" represents the cumulative number of pixels occupied by the sky components in the same image.

Green View Index

The Green View Index serves as a significant indicator for assessing the quality of greening. It quantifies the amount of green landscape visible within the human field of vision and reflects the composition of urban vertical space greening. After Aoki's proposal in [64], it garnered widespread adoption in subsequent research endeavors and yielded favorable outcomes [65–67]. The Green View Index was selected as one of the markers for quantifying the green space of the roadway. The calculation formula, Formula (11), is shown below:

$$GVI = \frac{Area_g}{Area_t} \tag{11}$$

The variable "$Area_g$" represents the total number of pixels occupied by green plant components in the segmented street view picture, and "$Area_t$" denotes the overall number of pixels in the image.

### 2.4. Data Analysis

The random forest technique was chosen in this study to construct a model for assessing the evaluation index and emotional perception of street greening. The random forest method is an ensemble learning technique that is founded upon the principles of decision trees. The system has the capability to simultaneously train datasets utilizing multiple decision trees. The initial dataset can be partitioned into many subsets using bootstrap sampling, with each subset serving as training data for constructing new decision trees and conducting subsequent analyses. After the prediction results were summarized, the average of the prediction results of all decision trees was finally output [68].

The Shapley Additive Explanation (SHAP) technique was selected as the approach to measure the impact of greening indicators on emotional perception. SHAP is a tool for interpreting the output of machine learning models derived from game theory and its related extension Shapley value concept [69]. The SHAP approach, when applied to a machine learning model, enables the elucidation of the impact of individual features on the projected value. Additionally, it provides insights into whether the influence is positive or negative. This study employed the SHAP method to allocate Shapley values to individual

elements based on the impact of several indicators of street greening on the perception of emotions. The formula for the SHAP approach is represented by Formula (12).

$$g(z') = \phi_0 + \sum_{i=1}^{M} \phi_i z_i' \tag{12}$$

The variable $Z_i'$, where $Z_i' \in \{0,1\}^M$, represents the participation of the $i$-th greening index in the model prediction. The variable $M$ represents the number of features in the perception model, and $\varphi_i$ indicates the impact value associated with the $i$-th greening index. The character $\varphi_0$ denotes the average value of the explained variable. The formula for calculating the Shapley value, as stated in reference [69], is represented as Formula (13):

$$\phi_i = \sum_{S \subseteq F \setminus \{i\}} \frac{|S|!(|F| - |S| - 1)!}{|F|!} \left[ f_{S \cup \{i\}} \left( x_{S \cup \{i\}} \right) - f_S(x_s) \right] \tag{13}$$

The symbol $\varphi_i$ denotes the Shapley value of the $i$-th greening index. The set $S$ represents the collection of greening indicators participating in the prediction. The variable $f$ represents the emotional perception model. In our work, the Shapley value represents the weighted summation of the difference in emotional perception scores of different scenes, and it accounts for the combination of scores of different greening indicators. The Shapley value represents the direct impact of street greening indexes on people's emotional perception. The magnitude of a value directly correlates with the extent of its influence. Additionally, the numerical value of SHAP also signifies the nature of its impact, whether it is positive or negative.

## 3. Results

### 3.1. The Holistic Influence of Greening Indicators on Emotional Perception

The research findings shown in Figure 6 demonstrate that the average SHAP values for the Green View Index (GVI) and Plant Level Diversity (PLD) are 0.208 and 0.225, respectively. These two greening indicators have the most significant influence on human emotional perception. Figure 7 shows that from the standpoint of positive emotions represented by pleasure and relaxation, the impact range of GVI on pleasure reached {−1.2~1.5}, and the impact range of PLD on relaxation reached {−1~0.8}. These findings suggest that GVI and PLD are the primary factors influencing the perception of positive emotions. Higher values of GVI and PLD correspond to a stronger positive impact on positive emotion perception, while lower values have a greater negative impact on positive emotions. From an alternative standpoint, when considering negative emotions such as boredom and anxiety, the impact range of GVI on boredom reached {−1.2~0.9}, and the impact range of PLD on anxiety reached {−0.6~0.8}. This indicates that GVI and PLD are also the two indicators that have the greatest impact on the perception of negative emotions, and the higher the values of GVI and PLD, the greater the negative impact on negative emotions. The lower their values, the greater their positive impact on negative emotions. The Simpson index, a metric used to quantify the level of equilibrium within a plant community, does not exhibit any substantial influence on emotional experience. Then, we selected several indicators that had a more significant impact on emotional perception for further clarification.

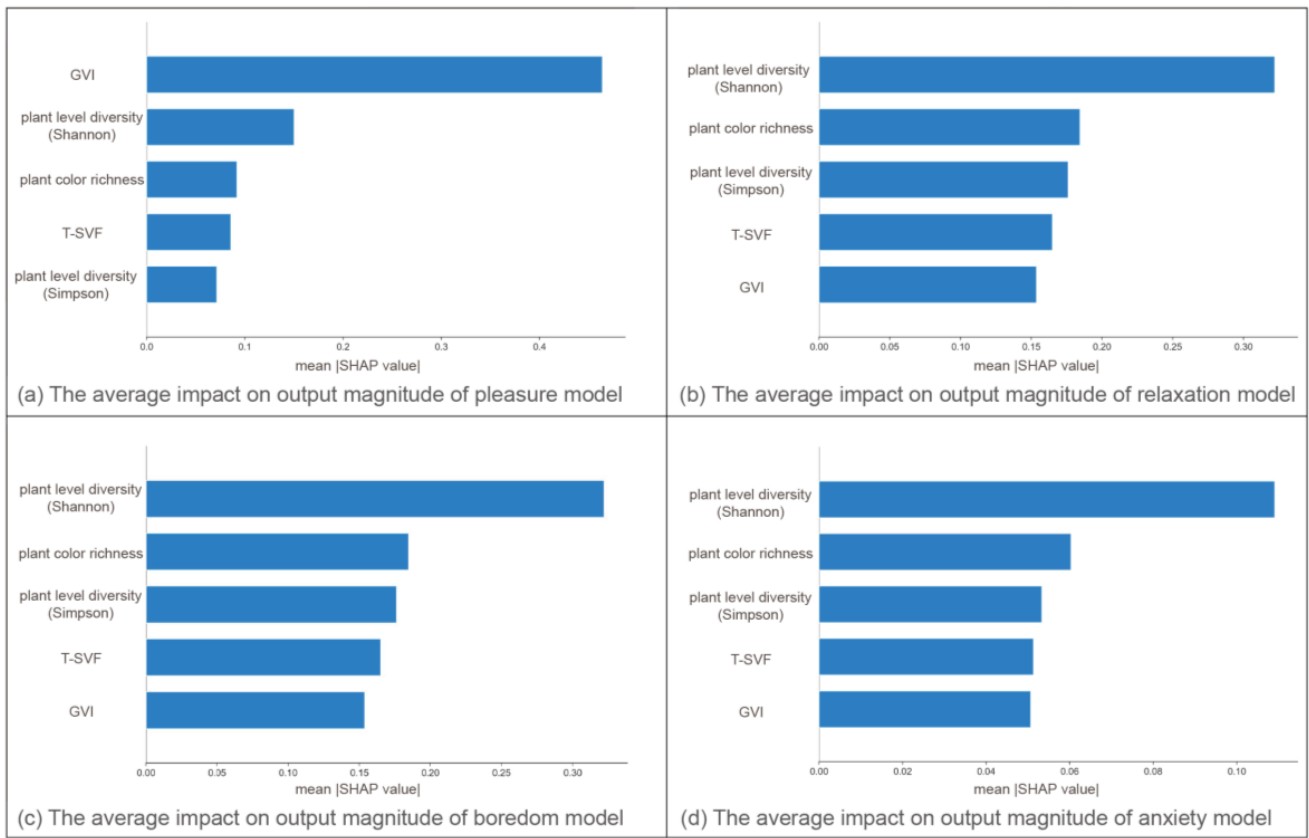

**Figure 6.** The graphs display the mean SHAP value of each greening index for the four emotional perceptions. The mean SHAP value of a feature indicates the magnitude of the feature's average impact on the model's output throughout the whole dataset. A high mean SHAP value indicates that the feature has a significant impact on the overall model output.

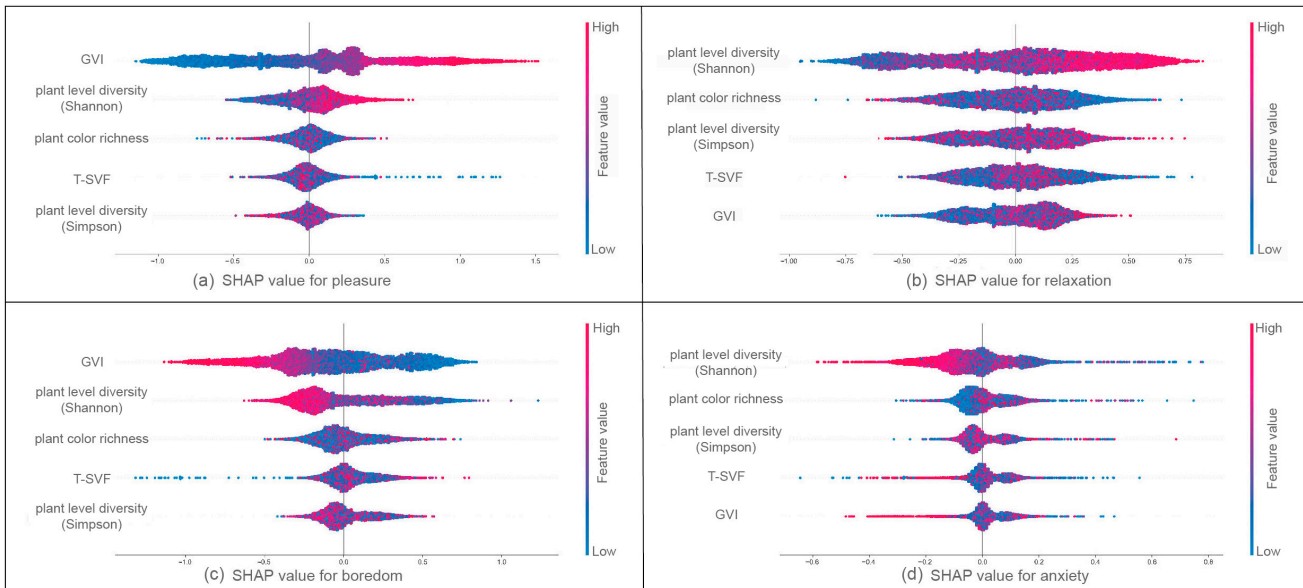

**Figure 7.** The graphs display the comprehensive influence of greening indicators on emotional perception. The positioning of the labels on the left side of each image depends on how much influence each indication exhibits. The Shapley value on the horizontal axis below represents the weight of the influence associated with each indicator. In the graph shown, each individual dot symbolizes a sample, with the color of the dot indicating the numerical magnitude of the respective sample. Bluer colors have lower values, whereas redder colors have higher values.

*3.2. The Influence of Typical Greening Indicators on Emotional Perception*

3.2.1. The Degree and Direction of Influence of Plant Level Diversity (PLD)

Figure 8a–d describe the changes in SHAP values with respect to the augmentation of PLD indicators. As observed in Figure 8a,b, as PLD increases, pleasure and relaxation also show an increasing trend, and the trend of increased relaxation is greater than that of pleasure. When PLD exceeds 0.9, its impact on positive emotions is almost entirely positive. Nevertheless, it is worth noting that a small portion (0.1%) of samples with high PLD correspond to lower positive emotions. Figure 8c,d show a decreasing trend in boredom and anxiety as PLD increases. When the level of PLD is below 0.2, negative emotion is almost promoted; however, when the level of PLD exceeds 0.4, it initiates the manifestation of inhibitory effects on negative emotions. Furthermore, when PLD surpasses 0.9, the impact on negative feelings is predominantly inhibitory. According to the findings shown in Figure 7, PLD exerts a more suppressive influence on negative emotions compared to its facilitative impact on happy emotions.

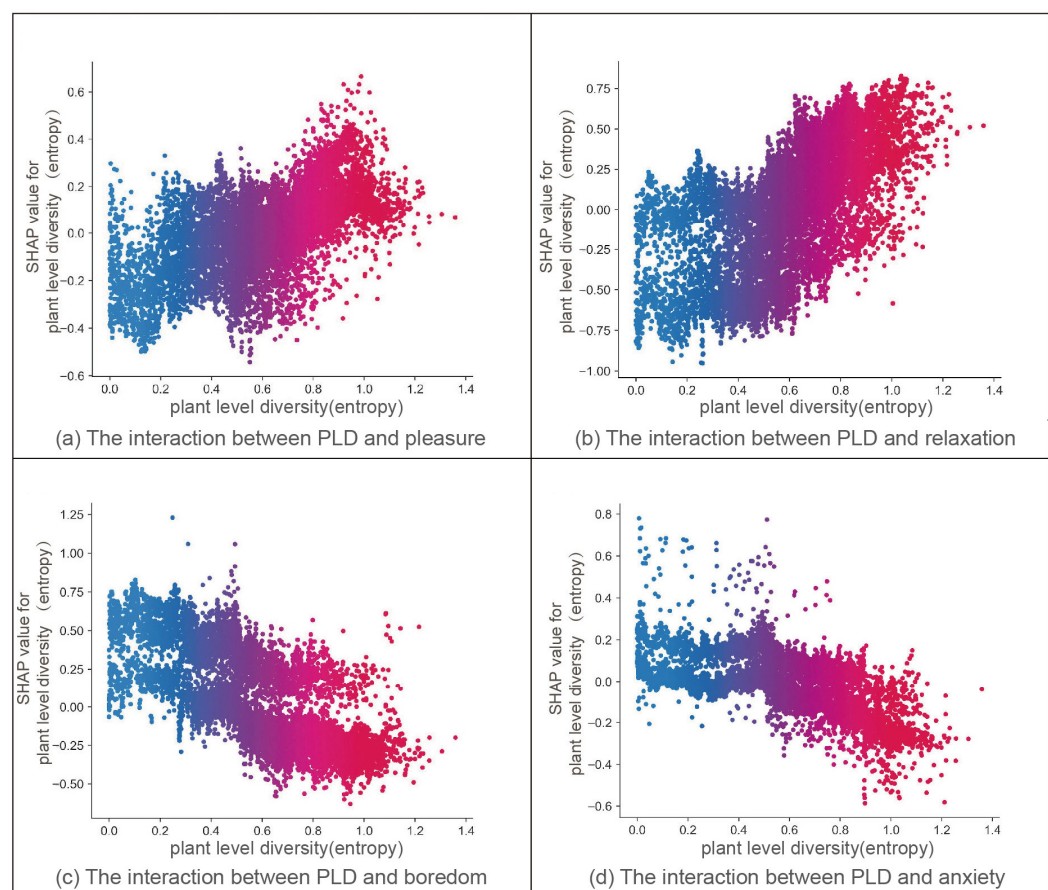

**Figure 8.** The impact of Plant Level Diversity (PLD) on SHAP values of different emotional perceptions.

3.2.2. The Degree and Direction of Influence of the Green View Index (GVI)

Figure 9a–d indicate the alterations in SHAP values as GVI increases. The link between GVI and pleasure is almost linear in Figure 9a, with a higher GVI having a stronger positive influence on pleasure. The data shown in Figure 9b suggest that when GVI is less than 0.5, the trend of influence on relaxation is not significant. However, when the GVI is above a threshold of 0.5, it has a more pronounced beneficial influence on relaxation. Importantly, a minor fraction (0.16%) of high GVI samples exhibit a negative impact on relaxation. In relation to negative feelings, Figure 9c illustrates that as GVI increases, the overall trend of boredom decreases. Moreover, it is observed that the suppressive impact of a high GVI

on feelings of boredom is considerably more pronounced than the stimulating effect of a low GVI on boredom. The data presented in Figure 9d show that when GVI is below 0.5, it has a promoting effect on anxiety, but the trend of change is not significant enough. However, as GVI increases to approximately 0.6, it starts to have a substantial inhibitory impact on anxiety. Based on Figure 7, it is evident that GVI has a stronger propensity to enhance positive emotions than to inhibit negative emotions.

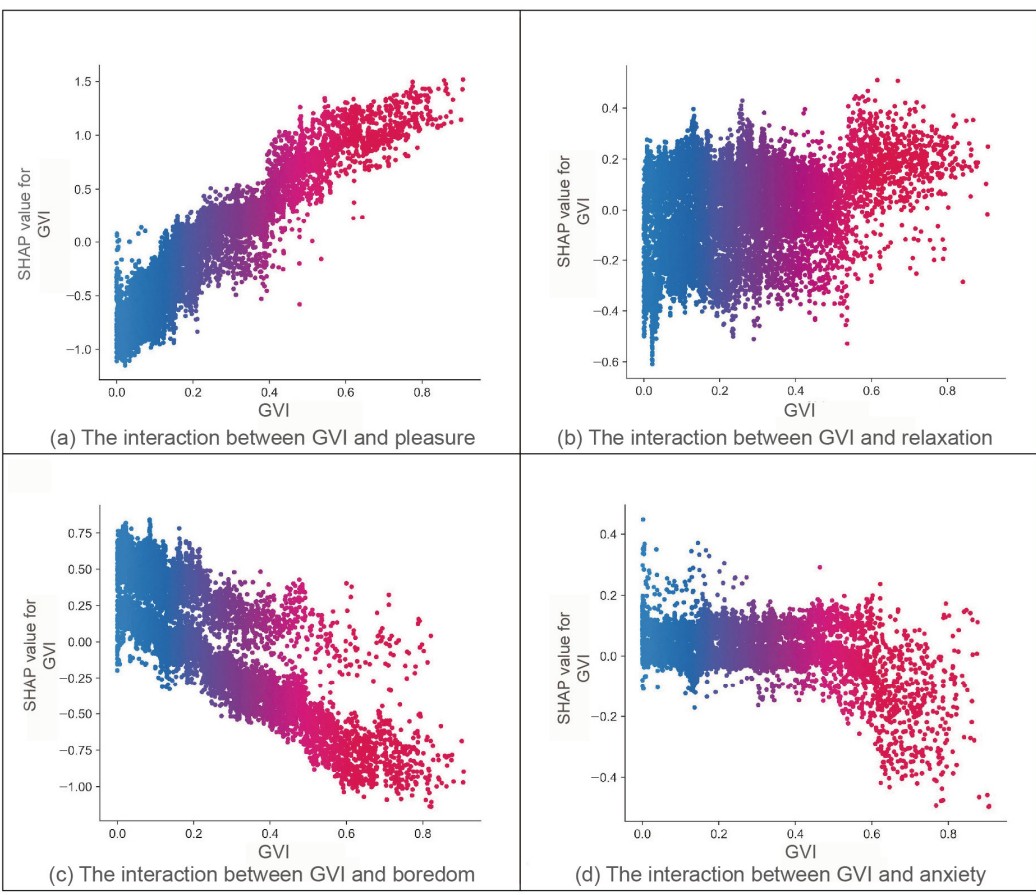

**Figure 9.** The impact of Green View Index (GVI) on SHAP values of different emotional perceptions.

### 3.2.3. The Degree and Direction of Influence of the Tree–Sky View Factor (T-SVF)

Figure 10a–d describe the quantitative impact of T-SVF on emotional perception. Figure 10a–d reveal that when the T-SVF increases, there is a fluctuation in the SHAP values of pleasure and relaxation, indicating a slightly decreasing–increasing–decreasing pattern in pleasant emotions. When the T-SVF is approximately 0.3, it exhibits its highest negative influence on positive emotions. In addition, as the T-SVF increases to approximately 0.4, it demonstrates its greatest positive impact on pleasure perception. Furthermore, when the T-SVF continues to rise to approximately 0.5–0.6, it exerts its most significant positive influence on relaxation. However, as it continues to increase, the SHAP value of positive emotional perception begins to decline again. In terms of negative emotions, the T-SVF has the most negative effect on the experience of anxiety when it falls within the range of 0.5 to 0.6. Nevertheless, its influence on the feeling of boredom is not deemed statistically significant.

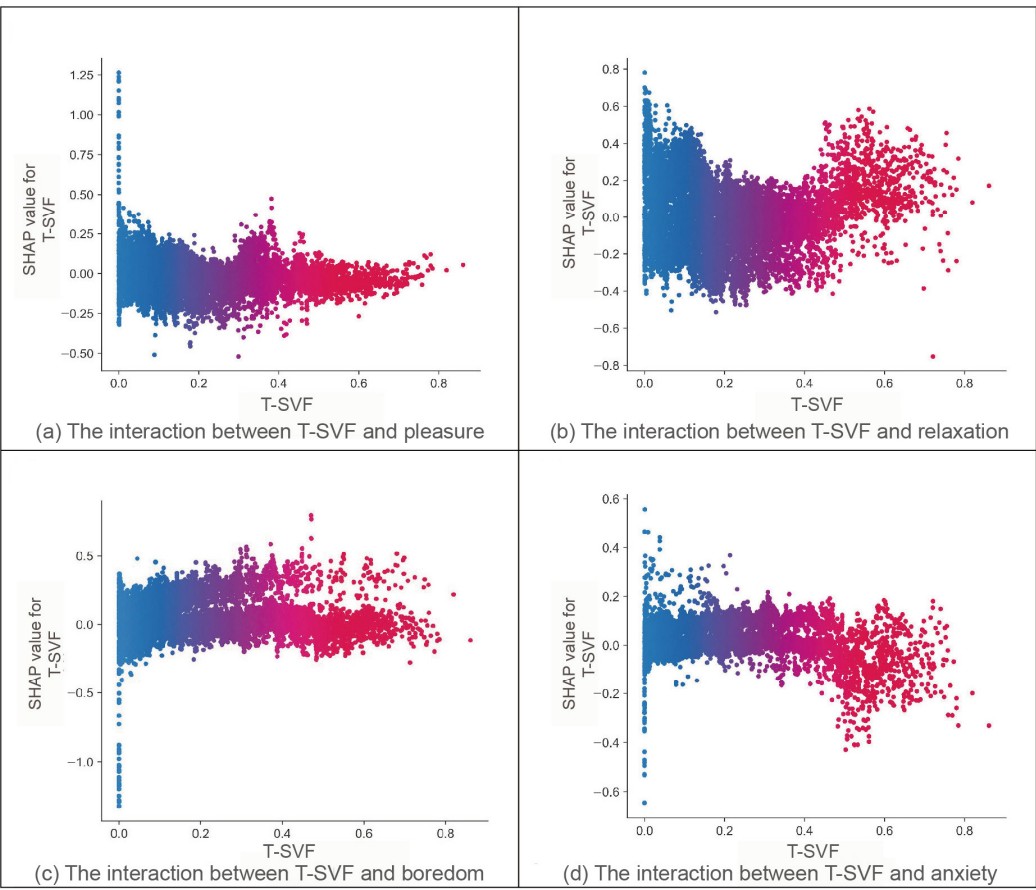

**Figure 10.** The impact of Tree–Sky View Factor (T-SVF) on SHAP values of different emotional perceptions.

### 3.2.4. The Degree and Direction of Influence of Plant Color Richness (PCR)

Figure 11a–d illustrate PCR's impact on emotion perception quantitatively. The trend of SHAP value change with increasing PCR, depicted in Figure 11a–d, is less pronounced compared to the indicators mentioned earlier. In Figure 11a, data reveal that high-PCR samples exhibit both positive and negative SHAP values, suggesting that increased PCR may have both positive and negative effects on pleasure perception. Figure 11b indicates that when PCR exceeds 15, its impact on relaxation falls within the range of {−0.6~0.2}. These samples are concentrated in the negative SHAP value area, suggesting a potential inhibitory relationship between high color richness and relaxation. Regarding negative emotions, Figure 11c shows that samples with a PCR range of 0–10 are concentrated in the positive SHAP value area, while those with a PCR range of 15–30 are concentrated in the negative SHAP value region. This indicates that low PCR promotes boredom perception, while high PCR suppresses it. Figure 11d suggests no statistically significant relationship between PCR and anxiety experience.

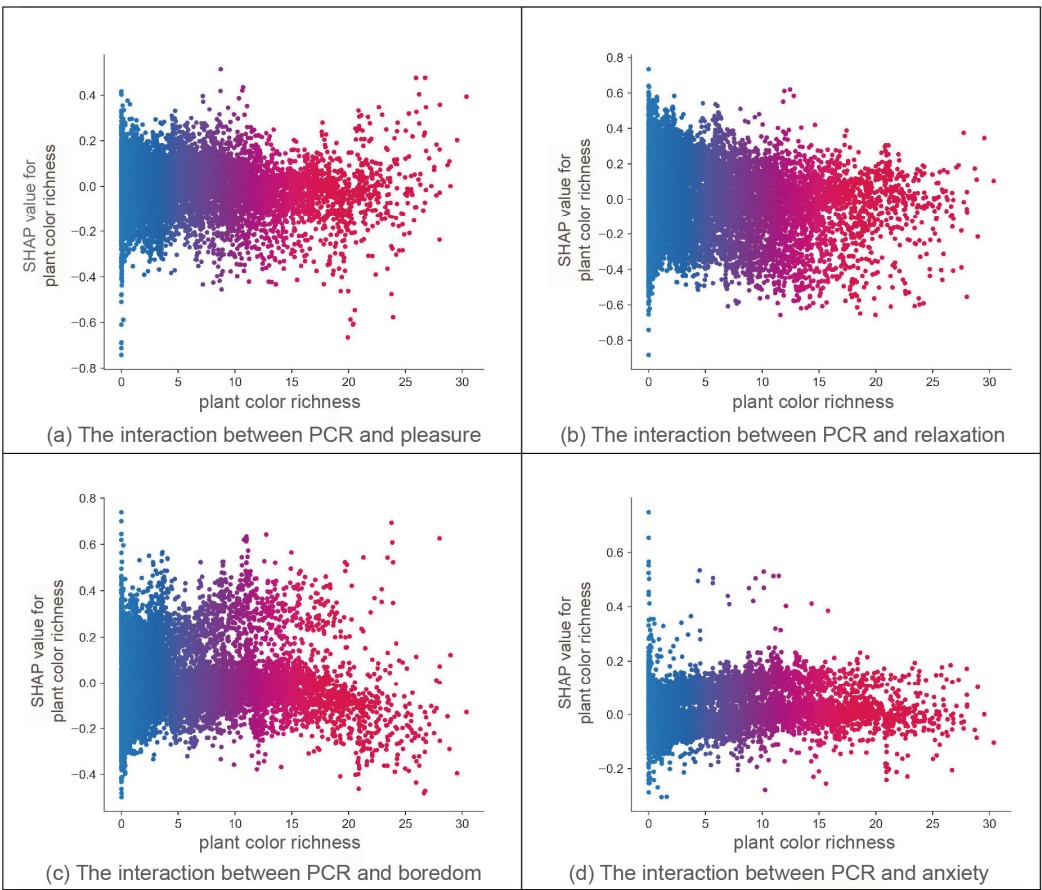

**Figure 11.** The impact of Plant Color Richness (PCR) on SHAP values of different emotional perceptions.

## 4. Discussion

### *4.1. Research Findings and Discussion*

This study used the random forest model and SHAP interpretation framework to examine emotional perception assessment and affecting factors in three major urban districts in Hangzhou, China. The quantification of street greening quality through the segmentation and calculation of street view images served as an explanatory variable in the model. Meanwhile, we employed street view big data and machine learning technology to create a dataset for emotional perception, which was used as the dependent variable in the model. Subsequently, the random forest algorithm was utilized to establish a green perception model, followed by the incorporation of the SHAP approach for the purpose of conducting an explanatory analysis on the aforementioned model. We identified key green indicators that affect emotional perception and conducted an in-depth analysis of the impact mechanism of various indicators of street greening on human emotional perception based on the SHAP value.

The applicability of our method extends beyond Hangzhou, China, and possesses a certain degree of universality. The cause of this phenomenon can be attributed to the practicality of semantic segmentation technology and the possibility of creating a dataset for emotion perception in diverse urban settings. This study employed semantic segmentation technology to precisely extract the plant components in street view images. This approach is applicable not only to locations with abundant greenery but also efficiently captures the intricate aspects of vegetation in less-green areas. Therefore, the technique is not constrained by the extent of vegetation growth and offers a viable approach for evaluating greening in diverse settings. Furthermore, our study integrated the emotional perception dataset created through manual scoring and machine learning to comprehensively account

for differences in individual subjective experiences. This enables our method to accurately capture individuals' emotional encounters in various urban environments, thereby providing urban planners with a more thorough understanding of emotional perception.

### 4.1.1. Analysis of the Impact of the Plant Level Diversity (PLD) on Emotional Perception

Our research findings indicate that when PLD is low, the plant community in the image is relatively uniform, which may be associated with a diminished aesthetic experience. The level of pleasure experienced by the viewer is relatively low, while the levels of boredom and worry are comparatively high. This scenario is depicted in Figure 12. The augmentation of plant variety levels elicits a favorable influence on the observer's subjective experience of pleasure and relaxation while concurrently engendering a negative impact on the feeling of boredom. The observed phenomenon might perhaps be attributed to the combined impact of floral communities, shrub communities, and tree communities exhibited in the images [70]. Figure 13 illustrates a situation with high Plant Level Diversity. Prior research has indicated that flowers have the capacity to elicit an obvious visual allure and stimulation, hence contributing to the augmentation of an individual's sense of pleasure and enrichment of their aesthetic experience [71]. The presence of shrubs and herbaceous plant types can also provide favorable effects on enjoyment [72]. Simultaneously, a green background can also help the observer recover from negative emotions [71]. Nevertheless, our research findings also indicate that a small number of images with higher Plant Level Diversity correspond to lower levels of pleasure and higher levels of anxiety. This phenomenon may be attributed to the observation that the plant community forms depicted in these images may be seen as excessively natural, leading to a sense of "chaos" and "disorder" among people viewing them. Figure 14 exemplifies this scenario. According to a study conducted by Qiu, Lindberg, and Nielson, it has been observed that human-modified plant landscapes, consisting of features such as clipped plant edges and lawns, tend to be more preferred by people than plant communities exhibiting the highest levels of biological variety [73]. Hence, it is imperative to thoroughly assess and regulate the natural extent and structure of plant communities.

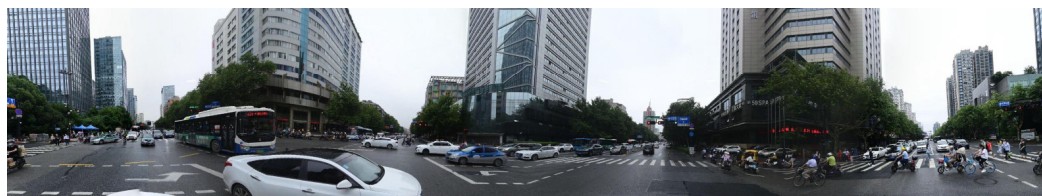

**Figure 12.** Example of low Plant Level Diversity (PLD). The associated emotional perceptions scored are as follows: Pleasure (2), Relaxation (1), Boredom (4), and Anxiety (3).

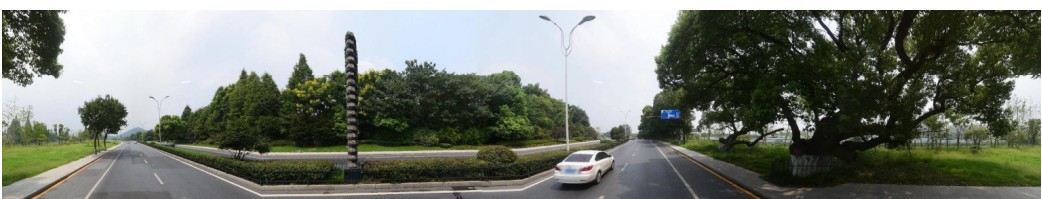

**Figure 13.** Example of high Plant Level Diversity (PLD). The associated emotional perceptions scored are as follows: Pleasure (4), Relaxation (4), Boredom (1), and Anxiety (2).

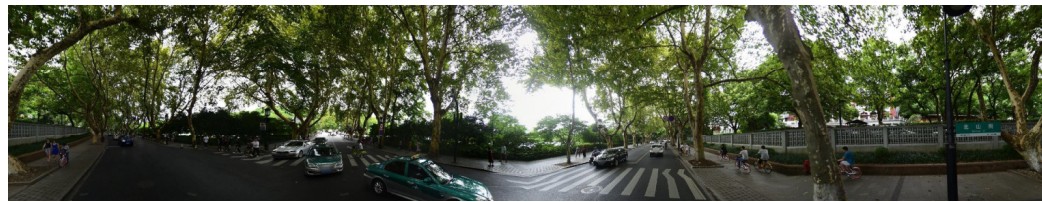

**Figure 14.** Example of disorder scenario. The associated emotional perceptions scored are as follows: Pleasure (2), Relaxation (3), Boredom (3), and Anxiety (4).

### 4.1.2. Analysis of the Impact of Green View Index (GVI) on Emotional Perception

Similar to plant landscape-level diversity, GVI also exhibits a non-linear relationship with emotional perception. The data presented in this study show that as GVI gradually increases from a low level, it has a positive promoting influence on pleasure and relaxation and an inhibitory effect on anxiety and boredom. These findings align with the results drawn in prior research conducted by Hartley et al. [74]. Figure 15 illustrates a scenario with high GVI. Nevertheless, some studies have indicated that a persistently elevated GVI may lead to a decline in individuals' sense of pleasure. This is because excessively dense vegetation reduces the penetration of light and blocks the line of sight [75], thus evoking feelings of oppression and insecurity among individuals [76]. However, our research data did not show a similar pattern, suggesting that this discrepancy may be attributed to the spatial heterogeneity of various research areas. Our research region exhibits a notable level of socioeconomic development, a considerable degree of urbanization, and favorable social security conditions. These factors may contribute to the mitigation of insecurity arising from too-dense green space. In contrast to the present study, investigations conducted by other researchers may encompass certain underdeveloped areas. In addition, our research also shows that some low-GVI images correspond to a higher sense of relaxation. This discovery might be attributed to the phenomenon that after observing many high-GVI images, the randomly occurring low-GVI images quickly transform their visual threshold content from a closed forest to bright and open green spaces, which helps alleviate tense and fearful emotions. Thus, the individual's good feelings are likely to see rapid improvement within a brief timeframe. This finding suggests that individuals experience heightened positive emotions in an environment characterized by alternating high and low GVI, as opposed to a setting with a single GVI [77].

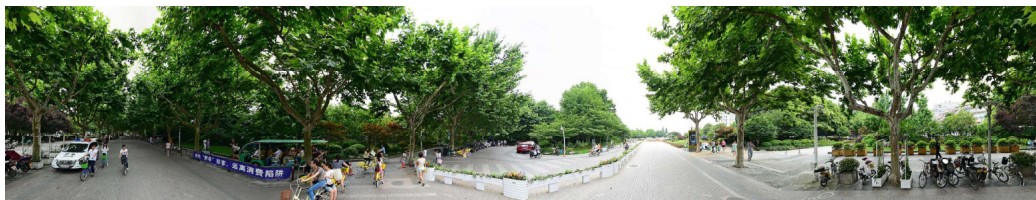

**Figure 15.** Example of high Green View Index (GVI). The associated emotional perceptions scored are as follows: Pleasure (5), Relaxation (4), Boredom (1), and Anxiety (2).

### 4.1.3. Analysis of the Impact of Tree–Sky View Factor (T-SVF) on Emotional Perception

In relation to the influence of T-SVF on emotional perception, our investigations indicate that within the range of 0.5 to 0.6, it has the most favorable effect on relaxation perception (with SHAP values ranging from 0.5 to 0.6). Conversely, it has a strong negative effect on the perception of anxiety (SHAP value interval {−0.4~−0.2}). This scenario is depicted in Figure 16. The observed effect might be caused by the fact that the T-SVF within this particular range signifies the most suitable ratio of trees to the sky, which ensures sufficient greenery to generate positive emotions, but the greenery is not too dense and does not affect the width of the view; it also guarantees that there is enough visible sky in the field of view, which has a significant effect on reducing the feeling of oppression [75]. When the T-SVF value is below 0.3, it implies a scarcity of trees in the image

and a relatively large proportion of sky. Such a scene might be too spacious to provide sufficient visual stimulation to the observer and cause boredom. Figure 17 exemplifies this scenario. The situation in which the T-SVF is excessively high (exceeds 0.6) shows similarity to the situation in which the GVI is excessively high, both of which might be attributed to dense plants blocking vision and causing insecurity, which is not conducive to people experiencing relief from anxiety. Figure 18 illustrates a situation with high T-SVF.

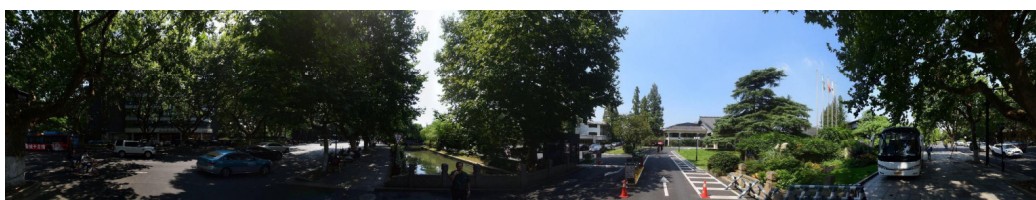

**Figure 16.** Example of Tree–Sky View Factor (T-SVF) values in the range of 0.5–0.6, highlighting a significant positive impact on enhancing the perception of relaxation. Conversely, it indicates a noticeable negative effect on anxiety perception within this range. The associated emotional perceptions scored are as follows: Pleasure (4), Relaxation (3), Boredom (2), and Anxiety (1).

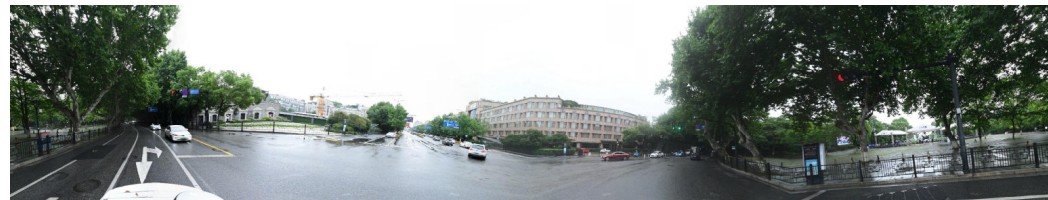

**Figure 17.** Example of low Tree–Sky View Factor (T-SVF). The associated emotional perceptions scored are as follows: Pleasure (2), Relaxation (4), Boredom (5), and Anxiety (2).

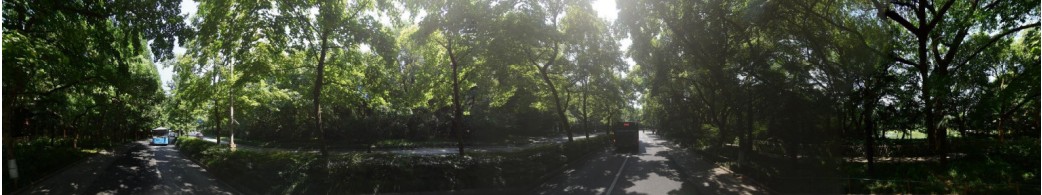

**Figure 18.** Example of high Tree–Sky View Factor (T-SVF). The associated emotional perceptions scored are as follows: Pleasure (4), Relaxation (2), Boredom (1), and Anxiety (3).

### 4.1.4. Analysis of the Impact of Plant Color Richness (PCR) on Emotional Perception

As mentioned in Section 3.2.4, our study indicates that the influence of PCR on emotional perception is not as significant as other indicators. Nonetheless, it is still possible to derive some inspiration from this component of the study. The information provided by Figure 10c confirms that samples with high PCR have an inhibitory effect on boredom perception, which might be attributed to the ability of colorful landscapes to better stimulate the visual function of observers and attract their attention [78]. According to the Attention Restorative Theory, when attention is attracted to the external landscape, the individual's directed attention will temporarily be restored, resulting in a reduction in negative affective states such as boredom. In terms of its influence on positive emotions, the evidence shown in Figure 10b indicates that samples with high PCR have a suppressive influence on the perception of relaxation. This finding could be attributed to the intense visual stimulation elicited by excessively vibrant and bright colors [79], which is not conducive to the brain's recovery from anxiety and relaxation. The conclusions of Paraskevopoulou et al. proved that an increase in Plant Color Richness has a positive impact on people's perception of pleasure [80], but our results are not entirely consistent with this. The results shown in this study indicate that samples exhibiting a high degree of color richness are found in regions

that both enhance and inhibit the feeling of pleasure. We believe that this discrepancy may be caused by subjective differences among individual participants. Individuals with heightened sensitivity to colors may encounter discomfort because of the impact exerted by intense colors on their vision [79]. Conversely, certain individuals may find themselves invigorated by vibrant and saturated colors, therefore experiencing a surge of positive emotions [81].

*4.2. Contribution to the Urban Street Greening Optimization Strategy*

Our research endeavors might serve as a source of support and analysis for local planning departments for decision making on urban planning and renewal oriented toward residents' well-being. To provide urban residents with a better green perception experience and higher emotional value, based on the research results of this study, we suggest the following: (1) In terms of the arrangement of plant communities at the street level, the proportion of flowers, ground cover plants, and shrubs can be appropriately increased to enrich the spatial level of plant communities without affecting visual permeability. The Hangzhou Boulevard Design Guidelines [82] proposed that the planting of trees on both sides of the road should be customized to suit various road functions and requirements. Additionally, enhancing the visual appeal of street plant landscapes involves enriching the hierarchy of plants while considering local conditions without compromising road safety. Furthermore, for areas with high negative emotional perception, we suggest moderate artificial intervention to the existing chaotic and disordered plant communities to make them more organized and improve people's perception of safety. (2) In the process of selecting plant species for street landscaping, in addition to green plants, plants with bright leaf colors and flowering plants could be considered when appropriate. However, considering that overly rich and vibrant colors are not conducive to people's recovery from anxiety, we recommend that street plants should be mainly based on green colors [83,84] paired with seasonal plants such as camellia, ginkgo, and goldenrain trees. The Implementation Opinions of the General Office of the Hangzhou Municipal People's Government on Scientific Greening in Hangzhou [85] also proposed the idea of constructing "colourful forests". It should also be noted that during the design phase, it is imperative to thoroughly consider the times of flowering and leaf color changes in various plant species, ensuring that the plants along the street exhibit a diverse range of colors throughout the year. (3) For built-up areas with low GVI, vertical greening, roof greening, and other forms are recommended to be used to increase the proportion of visible green areas. In contrast, in areas where the GVI is extremely high, appropriate pruning of existing plants is advisable to increase general visibility and sky visibility to reduce residents' sense of insecurity. We also suggest establishing plant landscapes with different GVI at intervals on the streets, as the alternating changes in high and low green rates can help stimulate people's positive emotions. (4) Considering the positive emotional effect of street greening, we suggest using evergreen plants like Camphor, Osmanthus fragrans, and pine to provide a consistent and pleasant emotional experience for people all year round. This advice aims to highlight that even in the winter season, when plants are not actively growing, certain evergreen species may nevertheless offer visual greenery and enduring mental benefits to inhabitants.

*4.3. Study Limitations*

Our research does possess certain limitations. First, although street view images have been proven to effectively connect the urban environment and human perception, some restrictions of using street view images cannot be ignored. The infrequent updating of street view images [86] limits their ability to accurately depict the seasonal changes in plants, particularly for herbaceous flowers and deciduous trees. Due to this limitation, street view images may impact our evaluation of the overall quality of street greening by only capturing the street environment at a specific moment in time [87]. Due to the fact that street view images are typically obtained from map vendors, it is not accessible to us to specify the weather conditions during the time of capturing the images, which could

partially affect PCR measurements. Furthermore, in addition to visual stimuli, human emotions can also be influenced by various environmental factors, such as temperature, weather conditions, and smells [88]. Future research should comprehensively assess these factors, in conjunction with advanced technologies such as scene 3D reconstruction [89], to thoroughly evaluate the influence of green environments on emotional perception from multiple perspectives throughout the entire period. Second, we collected fundamental data on emotional perception through volunteers scoring street view images. While efforts have been made to eliminate invalid samples, it should be acknowledged that there might still be some discrepancies between the experimental results and the actual situation. This phenomenon could be related to variations in cognition and cultural background [90]. Future research may require the development of a methodology that integrates questionnaire surveys with wearable physiological monitoring systems to assess emotional perception. For instance, by quantifying the emotional changes in the subjects with an electroencephalogram, one could enhance the scientific rigor of the investigation [91]. Third, although the survey yielded valuable data and underwent meticulous examination and discussion of the research findings, its scope was constrained by a restricted number of participants. Hence, the extent to which the findings can be applied to other contexts may be rather limited. Future research could comprehensively examine the influence of roadside vegetation on individuals' emotional perception by increasing the sample size and diversity.

## 5. Conclusions

In this study, we implemented a multi-dimensional measuring approach to assess the street greening in Hangzhou during the spring, summer, and fall seasons. We quantitatively assessed the extent of street greening and created a model using the SHAP algorithm to analyze how street greening affects emotional perception. Additionally, we conducted a quantitative assessment to determine the effect of street greening on human emotions. Furthermore, we investigated the non-linear correlation between various indicators of greening and emotional perception. This study addresses the issue of insufficient explanatory power of existing urban perception models for the complex relationship between the green environment and emotional perception, and it is of great significance for the planning and updating of urban public spaces that focus on mental health.

The research results show that the impact of various greening indicators on emotions is non-linear. When Plant Level Diversity (PLD) has a value greater than 0.9 and the Green View Index (GVI) has a value greater than 0.6, it shows a strong promoting effect on positive emotions and a restraining effect on negative emotions. However, higher values of PLD and GVI may not necessarily indicate better outcomes. The impact of the Tree–Sky View Factor (T-SVF) and Plant Color Richness (PCR) needs to be analyzed specifically based on different emotional dimensions. Thus, it is recommended that urban planners and policy-making departments take proactive measures to establish relevant street design guidelines. The emphasis is on intervening at the street greening level to enhance plant levels and increasing the proportion of visible green areas in necessary locations. They should also appropriately enhance plant colors. In addition, it is recommended that they appropriately cultivate evergreen vegetation to guarantee a consistent presence of greenery throughout all seasons. In areas with low safety awareness, it is necessary to trim plants appropriately to improve general visibility and sky visibility. These design standards will facilitate the creation of enjoyable street settings, aid inhabitants in cultivating positive emotions and recuperating from bad emotions, and offer improved living spaces for urban dwellers.

**Author Contributions:** N.H. designed the study and methodology. N.H. and D.H. collected and organized the street view images. X.L. and W.N. developed the deep learning model. N.H. carried out the data analysis and wrote the manuscript. W.N. and D.H. revised the manuscript. All authors have read and agreed to the published version of the manuscript.

**Funding:** This research was financially supported by Zhejiang A&F University Scientific Research Development Fund Project under Grant No. 2023LFR120.

**Data Availability Statement:** Data are contained within the article.

**Conflicts of Interest:** The authors declare no conflicts of interest.

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
