# Peer review of "Quantifying the Impact of Street Greening during Full-Leaf Seasons on Emotional Perception: Guidelines for Resident Well-Being"

_forests, doi:10.3390/f15010119_

Round 1
Reviewer 1 Report
Comments and Suggestions for Authors
In this paper, the authors employed various methods to explore how street greening influences people’s emotional perceptions. The paper is very well-written and it was a pleasure to read it. I praise the authors for using such a complex and effective methodology. I also liked the Discussion section in which the authors clearly demonstrated the value added of the research compared to the existing ones.
I only found the small number of participants having been asked to evaluate the street view images to be a weakness of the research; however, I understand how challenging it might have been to recruit 82 people to scrutinize so many images. I think this should be indicated as a limitation of the study.
Additionally, I recommend that the sentence in lines 166-169, on page 4, should be removed as it is a duplicate of the previous sentence.
Overall, I think this a high-quality paper.
Author Response
Dear Reviewer,
Thank you very much for your comments and recommendations. We have carefully evaluated and included your suggestions in the paper. Kindly see the included paper for comprehensive modifications implemented in response to your feedback.
Replies to reviewer 1’s comments:
In this paper, the authors employed various methods to explore how street greening influences people’s emotional perceptions. The paper is very well-written and it was a pleasure to read it. I praise the authors for using such a complex and effective methodology. I also liked the Discussion section in which the authors clearly demonstrated the value added of the research compared to the existing ones.
Response: We are very grateful to the reviewer for the comments and recognition of our research. According to the comments, we have carefully revised the contents of the manuscript to make the study more convincing. The specific modifications are as follows:
- I only found the small number of participants having been asked to evaluate the street view images to be a weakness of the research; however, I understand how challenging it might have been to recruit 82 people to scrutinize so many images. I think this should be indicated as a limitation of the study.
Response: Thanks for the reviewer’s understanding and suggestion. As a result of limited time and resources, the evaluation of street view images involved just 82 participants. The small number of participants is indeed one of the limitations of this study, and in response to the reviewer's recommendations, we have incorporated this constraint into Discussion section (Page 19, Lines 684-689).
- Additionally, I recommend that the sentence in lines 166-169, on page 4, should be removed as it is a duplicate of the previous sentence.
Response: Thanks for the reviewer's careful review. We were really sorry for our careless mistake. According to the reviewer's suggestion, we have removed the duplicate sentence (Page 4-5, Lines 172 -176).
Finally, thanks again to the reviewer for the careful review and helpful comments on our article, and we hope our modification above can be approved by you.
Reviewer 2 Report
Comments and Suggestions for Authors
The work presents a high level of research. The topic of the article and the discussed issues are current due to the current changes taking place in cities and urbanized spaces. The article is constructed by the IMRAD method and includes an introduction, a description of the method and research materials, research results, and an extensive discussion. It requires ordering in individual elements.
1. The title of the article is extensive, indicating the issues discussed, it is worth considering shortening the title.
2. The summary includes an indication of the thesis and research objective, the method used, and the research results described. The purpose of the work should be indicated, it results from the content of the summary but is not clearly defined.
Extended summary - it is suggested to shorten the summary. Research results - it is suggested to describe them more synthetically (in a generalized way), without indicating the exact values of the research results. Elements describing current research on the perception of the environment (sky, buildings, plants, vehicles) could be included in the description of the research problem, as a justification for taking up the topic or a research gap.
3. Introduction. The structure of the introduction is logical and clear but requires minor corrections. The introduction includes a literature review and a description of the state of the research, and the research background is outlined. The way of perceiving space using popular methods was indicated, and the types of data collection methods and processing techniques were determined - global positioning system (GPS), geographic information system (GIS), field research, street photos, and other methods of collecting data on vegetation. The impact of street greening on humans is shown. The selected method was indicated and the choice was justified. Research problems were highlighted. The strengths and weaknesses of the methods used are indicated, and the description of the state of research is suggested to be supplemented and to show additional advantages beyond those indicated as the main one - the possibility of assessing greenness from the human level - for the selected method. The introduction should begin with a description of the purpose of the study and the expected effect and end with a clear indication of the research gap. The introduction lacks broader information about the support of the selected method by the expert group for programming the algorithm.
4. Research methods. The research area is described briefly, and it is suggested to highlight the choice of site as a reference example to show the possibilities of applying the method. In section 2.2. the description of the method and the method of conducting research were indicated. There was a reference to human-machine data transfer. The research process (section 2.3) is described clearly, and divided into subsections, which makes it easier to understand.
It is worth adding an explanation of the reason for choosing four specific emotions as indicators of emotional perception, this is to explain the influence of the external environment on human emotions without the need to study bibliographic items. It is also worth indicating the age range of survey participants, apart from the average age, in the research description. There is an assumption that the selection of respondents with a large age range or of a similar age could influence the obtained research results. The strengths and weaknesses of the method and errors in the adopted data processing method were indicated. The division into sub-items improves readability, but the equations regarding the course of research are suggested to be more closely related to the description.
5. Research results - Exhaustive analysis of the results, and division into sub-points support in the interpretation of the results.
6. Discussion. The division into subsections improves readability and shows the analysis of various aspects of the study - there is a reference to research methods and materials. The strong point of the conclusion is the description of the contribution to the City Street Greening Optimization Strategy. The limitations of the method used (4.3.) are indicated, and ways to improve the selected method are shown.
In the discussion, it is worth pointing out the possibilities of using the method for other purposes or using the method for areas with little greenery or greenery to varying degrees. Emphasize the universality of the method.
7. Conclusions - The validity of conducting research and the possibility of using research and methods to improve the greenness of streets and areas near the streets were indicated. The chapter lists the factors that had the greatest impact on the obtained result and indicates the factors determining the feeling of space. Possibilities of using the study in the work of planners and urban planners were indicated.
Only minor editing and some polishing are needed
Author Response
Dear Reviewer,
Thank you very much for your comments and recommendations. We have carefully evaluated and included your suggestions in the paper. Kindly see the included paper for comprehensive modifications implemented in response to your feedback.
Replies to reviewer2’s comments:
The work presents a high level of research. The topic of the article and the discussed issues are current due to the current changes taking place in cities and urbanized spaces. The article is constructed by the IMRAD method and includes an introduction, a description of the method and research materials, research results, and an extensive discussion. It requires ordering in individual elements.
Response: We feel great thanks for the professional review work on our study. As you are concerned, there are several issues that need to be improved. According to your insightful recommendations, we have made extensive modifications to the prior manuscript. The detailed modifications are listed below.
- The title of the article is extensive, indicating the issues discussed, it is worth considering shortening the title.
Response: Thanks to the reviewer for the thorough review and helpful suggestions. We shortened and adjusted the title based on the reviewer's suggestions and the opinions of other reviewers. The most up-to-date title is: " Quantifying the Impact of Street Greenery in Full Leaf on Emotional Perception: Guidelines for Resident Well-being." We attempted to make this new title more concise and clearer, while retaining key information about the main content of the paper to better meet academic requirements and intelligibility (Page 1, Lines 2-3).
- Abstract. The summary includes an indication of the thesis and research objective, the method used, and the research results described. The purpose of the work should be indicated, it results from the content of the summary but is not clearly defined.
Extended summary - it is suggested to shorten the summary. Research results - it is suggested to describe them more synthetically (in a generalized way), without indicating the exact values of the research results. Elements describing current research on the perception of the environment (sky, buildings, plants, vehicles) could be included in the description of the research problem, as a justification for taking up the topic or a research gap.
Response: We greatly appreciate the reviewer’s helpful feedback. In response to the reviewer's recommendations, we thoroughly examined and modified the abstract section to enhance its conciseness and rigor. The precise alterations are as follows:
- We clearly defined the research aims in the abstract, as suggested by the reviewer. (Page 1, Lines 14-17).
- Many thanks to the reviewer for the comprehensive review of our paper and valuable recommendations. We carefully evaluated every word and sentence in response to the reviewer's advice, and trimmed details of prior research and our research techniques to ensure that the Abstract section is more concise. Simultaneously, the description of the research purpose, the street greening optimization approach, the street view image shooting time, and other information are added to ensure that the summary is more comprehensive. We were able to ensure that the word count in the abstract is more compact than in the original text after making these changes. (Page 1, Lines 11-27).
- We have removed the mention of SHAP values of 0.225 and 0.265 from the abstract session in response to the feedback from this reviewer, as well as reviewer 3 and reviewer 4. This decision was made because these numbers may not hold significant meaning for individuals unfamiliar with the research methodology. Following the reviewer's recommendation, we decided against specifying the exact value of the findings and instead opted to describe them in a general way (Page 1, Lines 21-23).
- According to the reviewer's suggestion, we have summarized "sky, buildings, plants and vehicles", which describe the current research on environmental perception, as "all objects identified by semantic segmentation of street view images". We have then incorporated this summary into the research question's description (Page 1, Lines 12-13).
- Introduction. The structure of the introduction is logical and clear but requires minor corrections. The introduction includes a literature review and a description of the state of the research, and the research background is outlined. The way of perceiving space using popular methods was indicated, and the types of data collection methods and processing techniques were determined - global positioning system (GPS), geographic information system (GIS), field research, street photos, and other methods of collecting data on vegetation. The impact of street greening on humans is shown. The selected method was indicated and the choice was justified. Research problems were highlighted. The strengths and weaknesses of the methods used are indicated, and the description of the state of research is suggested to be supplemented and to show additional advantages beyond those indicated as the main one - the possibility of assessing greenness from the human level - for the selected method. The introduction should begin with a description of the purpose of the study and the expected effect and end with a clear indication of the research gap. The introduction lacks broader information about the support of the selected method by the expert group for programming the algorithm.
Response: Thank you for your meticulous review and insightful suggestions. After thorough consideration of your remarks, we have carefully implemented the required modifications individually, which are detailed below.
- As per the reviewer's recommendation, we included two additional advantages regarding the research methodology, in addition to the primary benefit of assessing street greening from a human standpoint. By utilizing street view images and computer technology, the measurement of street greening may be conducted with greater accuracy and efficiency. This advancement also allows for the expansion of street greening research from small-scale to larger-scale applications (Page 2, Lines 65-69).
- According to the reviewer's suggestion, we revised the Introduction section to start with a description of the research purpose and expected effects and emphasized the research gap at the end (Page 1, Lines 33-34; Page 3, Lines 125-127).
- Regarding the computer algorithms used for our study, we thoroughly examined the Introduction section in accordance with the reviewer's recommendations and discovered that we only provided a concise evaluation of the most significant SHAP approaches. We proceeded to study literature on the implementation of this algorithm in urban research to verify its viability in aiding our comprehension and analysis of the impact of street greening on people' emotions. (Page 3, Lines 103; Page 22, Lines 815-822).The recently included citations are as follows:
Rui, J. Exploring the association between the settlement environment and residents’ positive sentiments in urban villages and formal settlements in Shenzhen. Sustain. Cities Soc. 2023, 98, 104851. https://doi.org/10.1016/j.scs.2023.104851
Su, S.; Wang, Z.; Li, B.; Kang, M. Deciphering the influence of TOD on metro ridership: An integrated approach of extended node-place model and interpretable machine learning with planning implications. J. Transp. Geogr. 2022, 104, 103455. https://doi.org/10.1016/j.jtrangeo.2022.103455
Yin, H.; Xiao, R.; Fei, X.; Zhang, Z.; Gao, Z.; Wan, Y.; Tan, W.; Jiang, X.; Cao, W.; Guo, Y. Analyzing "Economy-Society-Environment" sustainability from the perspective of urban spatial structure: A case study of the Yangtze River Delta Urban Agglomeration. Sustain. Cities Soc. 2023, 96, 104691. https://doi.org/10.1016/j.scs.2023.104691
We have added a summary of the potential uses of DeepLabV3+ in the Method section of our paper on semantic segmentation. (Page 5, Lines 196-198; Page 22, Lines 829-834;). The recently included citations are as follows:
Xia, Y.; Yabuki, N.; Fukuda, T. Development of a system for assessing the quality of urban street-level greenery using street view images and deep learning. Urban For. Urban Green. 2021, 59, 126995. https://doi.org/10.1016/j.ufug.2021.126995
Gao, F.; Li, S.; Tan, Z.; Zhang, X.; Lai, Z.; Tan, Z. How is urban greenness spatially associated with dockless bike sharing usage on weekdays, weekends, and holidays. ISPRS Int. J. Geoinf. 2021, 10, 238. https://doi.org/10.3390/ijgi10040238
Wang, M.; Vermeulen, F. Life between buildings from a street view image: What do big data analytics reveal about neighbourhood organisational vitality. Urban Stud. 2021, 58, 3118–3139. https://doi.org/10.1177/0042098020957198
- Research Methods. The research area is described briefly, and it is suggested to highlight the choice of site as a reference example to show the possibilities of applying the method. In section 2.2. the description of the method and the method of conducting research were indicated. There was a reference to human-machine data transfer. The research process (section 2.3) is described clearly, and divided into subsections, which makes it easier to understand.
It is worth adding an explanation of the reason for choosing four specific emotions as indicators of emotional perception, this is to explain the influence of the external environment on human emotions without the need to study bibliographic items. It is also worth indicating the age range of survey participants, apart from the average age, in the research description. There is an assumption that the selection of respondents with a large age range or of a similar age could influence the obtained research results. The strengths and weaknesses of the method and errors in the adopted data processing method were indicated. The division into sub-items improves readability, but the equations regarding the course of research are suggested to be more closely related to the description.
Response: Thank you for your wise recommendations. we have carefully revised our paper to improve the precision the manuscript. The specific suggestions and their corresponding responses are listed below.
- According to the reviewer's suggestion, we have included the rationales behind selecting four particular emotions as indicators of emotion perception. This addition aims to elucidate the impact of the external environment on the perception of emotions (Page 6, Lines 229-234).
- In response to the reviewer's recommendation, we incorporated the age range of participants in the experiment and included the corresponding proportion for each age group in Table 1(Page 6, Lines 239; Page 7, Lines 261).
- Gratitude is expressed to the reviewer for the meticulous evaluation and valuable recommendations. We have made appropriate revisions based on the reviewers' comments to enhance the strong correlation between the equation description and the research procedure. We have highlighted the physical significance of each equation to provide a clearer explanation of the unique function of each equation in the study design (Page 8, Lines 284-293).
- Research Results. Exhaustive analysis of the results, and division into sub-points support in the interpretation of the results
Response: Thanks a bunch to you for your careful review and invaluable suggestions. Your expertise has been instrumental in the improvement of my research.
- Discussion. The division into subsections improves readability and shows the analysis of various aspects of the study - there is a reference to research methods and materials. The strong point of the conclusion is the description of the contribution to the City Street Greening Optimization Strategy. The limitations of the method used (4.3.) are indicated, and ways to improve the selected method are shown.
In the discussion, it is worth pointing out the possibilities of using the method for other purposes or using the method for areas with little greenery or greenery to varying degrees. Emphasize the universality of the method.
Response: We appreciate your assessment of our article and the helpful suggestions you provided. In response to the reviewer's recommendations on the universality of our research approach, we have incorporated more pertinent information in the discussion section to demonstrate the broad applicability of our research method and the proposed greening optimization plan. The content explores the practicality of using semantic segmentation technology and the possibility of creating emotional perception datasets in different urban settings. This study employs semantic segmentation technology to precisely extract plant components from street view images. This method is applicable not only to highly green places but also efficiently captures the fine-grained vegetation characteristics in less green areas. Hence, the method is not constrained by the extent of vegetation growth and offers a viable approach for evaluating vegetation coverage in various settings. Furthermore, our study integrated the emotional perception dataset created through manual scoring and machine learning to comprehensively account for variations in individual subjective experiences. Our technique is capable of accurately capturing individuals' emotional experiences across diverse urban environments (Page 14-15, Lines 496-509).
- Conclusions. The validity of conducting research and the possibility of using research and methods to improve the greenness of streets and areas near the streets were indicated. The chapter lists the factors that had the greatest impact on the obtained result and indicates the factors determining the feeling of space. Possibilities of using the study in the work of planners and urban planners were indicated.
Response: We express our deep gratitude to the reviewer for dedicating their time and exerting much effort in reviewing my manuscript. Your comprehensive review and insightful ideas have offered crucial advice and motivation for my study. With the reviewer's help, we acquired a more comprehensive comprehension of our work, which was important in enhancing its quality.
Finally, once again, we would like to express our heartfelt thanks to the reviewer for the good suggestion. We hope that the current revisions will be well received by the reviewer.

Reviewer 3 Report
Comments and Suggestions for Authors
“How does street greening influence human emotional perception? An interpretable perception evaluation and analysis based on street view images and machine learning”
Title: The title is misleading because the article doesn’t really address the “how.” That would involve a different study in which brain mechanisms are analyzed for physiological arousal, flight response, love, and other human emotions. Also, the study only looked at streets that are “green” so the analysis really is for certain periods throughout the year when trees are in full leaf. The same findings can’t be assigned to humans traveling on a street during a winter blizzard. The study has purposes of giving information to local planning departments for their decision-making about urban planning (lines 543 and 544) and to then “help residents generate positive emotions and recover from negative emotions.” (lines 620 and 621). If this information was in the title, more potential readers would want to read the article and implement the findings.
What about “Determination of the influences of streets in full leaf to guide urban planners’ designs that could result in residents’ positive emotions and recovery from negative emotions.”
With this title, you are being specific about what and about why. The second part of the existing title just indicated that you did a study…”An interpretable perception evaluation and analysis based on street view images and machine learning.” This reviewer believes the authors have more value in their article than producing a study to advance their careers. With climate change and the world being in a state of turmoil, the authors have lessons for urban planners and ways to heal residents.
Abstract: The authors need to be watchful to not have too many segue words that don’t carry the proper meaning. The first “However” on line 14 is fine following the first sentence. The word “Nevertheless” on line 16 does not belong there. The authors here could write, “Therefore” because you have identified the problem already. This strongly sets up the next sentence. On Line 17, the authors would start with “Our study…’ What is missing in the Abstract are your findings. The authors at the end say what their study addressed but this was discussed in the first sentence in the Abstract. The authors have done more than “offer useful information for future research.” We have climate change and negative mental states from all of the stress in the world and daily life. The authors have concrete findings throughout the paper to tell urban planners what to build. This should be in the Abstract because not everyone reads the entire article. The authors need to tell urban planners what to “do” in the abstract and underscore the “why.” The readers need to read what will help them emotionally so they can advocate for those elements on their streets. The streets in Hanzhou are indeed lovely and other cities need to have similar plantings. The authors need to give the reader evidence so people in other cities can duplicate the streets in Hangzhou. The lines 35-36 explain the “why” of this article.
For the Abstract in lines 20-25, the authors need to provide the findings in lay terms. Reading the emotional perception influence number of 0.225 and 0.265 is not sufficiently meaningful to someone who does not understand these numbers. The authors also in the Abstract need to tell the readers that they were studying streets when they were in full leaf. Street greening implies that streets can be green all year. The street view images were on clear days when the trees were in full leaf and, with the plant color richness (PCR), the days would be sunny and not grey or rainy.
Introduction: The authors on line 76 tell the reader they will discuss two issues but the second issue is buried in the paragraph. “First” on line 78 starts a paragraph. “Second” on line 96 should start a new paragraph to help the reader.
The authors on lines 118 through 124 make statements but the article itself falls short. These sentences are overstatements. The findings also need to be clearer and more closely duplicate the work of the Kaplans (citation 44 line731-732). In the Kaplan’s many studies, they showed pictures of what was preferred. (Jumping ahead to the images Figures 6, 7, 8, 9, and 10, those figures in isolation have little meaning. It would be better to have a picture of a street in full leaf and then show the emotion related to that picture. It is understandable that the graphs are responses to multiple data sets, so perhaps several examples could be shown of what street views resulted in pleasure, relaxation, boredom, or anxiety.)
Methods: On lines 169-171, the authors discuss the four images used to create a panoramic picture. All of the pictures show trees in full leaf. Somewhere in the text the authors need to indicate that they chose images during specific months. Also, it is unusual to have panoramic images because a person can only look forward. It is possible on a street to have a positive view in one direction and a negative view in the opposite direction. It is rare to have a panoramic view that is similar all the way around. The authors need to discuss why they chose to use panoramic images because people only have eyes on one side of their head.
Discussion. This section would be helped by having pictures of what a viewer might have seen and the related emotional response. On lines 487-491, the authors discuss having too much greenness and reducing the penetration of light. In the images the authors showed, sky was always visible so there would not be a perception of oppressiveness because the viewer could see far.
Conclusion: The authors could look at the points written for the Abstract and provide more information. The authors might offer guidance about what to have in the off season for plants when the streets cannot be fully green and in full leaf. Resident needs to have emotional well being 12 months out of the year and not just when it is warm and sunny. While this could be in the Limitations section, the reader would be more appreciative if the authors acknowledge the need to help residents all year. There are findings for this in the data.
Author Response
Dear Reviewer,
Thank you very much for your comments and recommendations. We have carefully evaluated and included your suggestions in the paper. Kindly see the included paper for comprehensive modifications implemented in response to your feedback.
Replies to reviewer3’s comments:
- Tittle. The title is misleading because the article doesn’t really address the “how.” That would involve a different study in which brain mechanisms are analyzed for physiological arousal, flight response, love, and other human emotions. Also, the study only looked at streets that are “green” so the analysis really is for certain periods throughout the year when trees are in full leaf. The same findings can’t be assigned to humans traveling on a street during a winter blizzard. The study has purposes of giving information to local planning departments for their decision-making about urban planning (lines 543 and 544) and to then “help residents generate positive emotions and recover from negative emotions.” (lines 620 and 621). If this information was in the title, more potential readers would want to read the article and implement the findings.
What about “Determination of the influences of streets in full leaf to guide urban planners’ designs that could result in residents’ positive emotions and recovery from negative emotions.”
With this title, you are being specific about what and about why. The second part of the existing title just indicated that you did a study…” An interpretable perception evaluation and analysis based on street view images and machine learning.” This reviewer believes the authors have more value in their article than producing a study to advance their careers. With climate change and the world being in a state of turmoil, the authors have lessons for urban planners and ways to heal residents.
Response: Thanks to the reviewer for the insightful recommendations regarding the title of our study. After careful consideration of these comments we have made the necessary adjustments to ensure that the title is both precise and scholarly. The current title is: " Quantifying the Impact of Street Greenery in Full Leaf on Emotional Perception: Guidelines for Resident Well-being." The revised title emphasizes that our study quantifies emotional perception over the full leaf period of street greening, and provides clear suggestions for improving the mood and well-being of residents. This modification is founded on the recommendation of the reviewer, and we are of the opinion that it more accurately represents the essence and scholarly worth of the research (Page 1, Lines 2-3).
- Abstract. The authors need to be watchful to not have too many segue words that don’t carry the proper meaning. The first “However” on line 14 is fine following the first sentence. The word “Nevertheless” on line 16 does not belong there. The authors here could write, “Therefore” because you have identified the problem already. This strongly sets up the next sentence. On Line 17, the authors would start with “Our study…’ What is missing in the Abstract are your findings. The authors at the end say what their study addressed but this was discussed in the first sentence in the Abstract. The authors have done more than “offer useful information for future research.” We have climate change and negative mental states from all of the stress in the world and daily life. The authors have concrete findings throughout the paper to tell urban planners what to build. This should be in the Abstract because not everyone reads the entire article. The authors need to tell urban planners what to “do” in the abstract and underscore the “why.” The readers need to read what will help them emotionally so they can advocate for those elements on their streets. The streets in Hangzhou are indeed lovely and other cities need to have similar plantings. The authors need to give the reader evidence so people in other cities can duplicate the streets in Hangzhou. The lines 35-36 explain the “why” of this article.
For the Abstract in lines 20-25, the authors need to provide the findings in lay terms. Reading the emotional perception influence number of 0.225 and 0.265 is not sufficiently meaningful to someone who does not understand these numbers. The authors also in the Abstract need to tell the readers that they were studying streets when they were in full leaf. Street greening implies that streets can be green all year. The street view images were on clear days when the trees were in full leaf and, with the plant color richness (PCR), the days would be sunny and not grey or rainy.
Response: Thank you for your comprehensive evaluation of our work and the valuable recommendations offered. According to the comments, we have carefully revised our paper to improve the precision and clarity of the manuscript. The general comments and specific suggestions and their corresponding responses are listed below, and we sincerely hope that the revised manuscript will fulfill the high standards for publication.
- We carefully reviewed the meaning of segue words and the logical relationship between sentences in Abstract, as suggested by the reviewer, to ensure that segue words are utilized correctly and appropriately. As per the reviewer's recommendation, we substituted "Nevertheless" with "Therefore" in the original manuscript. Additionally, we eliminated "Therefore" and "However" from the original manuscript (Page 1, Lines 16).
- As suggested by the reviewers, we have incorporated sentences into Abstract to describe our particular findings and tell the city planners what to do and why (Pagec1, Lines 24-27).
- We have removed the mention of SHAP values of 0.225 and 0.265 from the abstract session in response to the feedback from this reviewer, as well as reviewer 2 and reviewer 4. This decision was made because these numbers may not hold significant meaning for individuals unfamiliar with the research methodology. Following the reviewer's recommendation, we decided against specifying the exact value of the findings and instead opted to describe them in a general way (Page 1, Lines 21-23).
- We apologize for missing the information about when these Street View images were captured while writing. In response to the reviewer's reminder, we have included in Abstract the fact that our research was conducted during the period when the plants along the street were in full leaves. Additionally, we have specified the time at which the street view photos were captured and mentioned that we have excluded or substituted any winter photos in the methodology chapter (Page 1, Lines 17-18; Page 4, Lines 169-172).
- Regarding the weather conditions mentioned by the reviewer during the capture of Street View images, we regret that we cannot ensure that all the Street View images we used were taken under sunny weather conditions. Because we're not the ones who took the street view images. The Street View images utilized in our research are sourced from Baidu Map. The Baidu Map supplier is the actual photographer of these images, and we cannot assign the weather at the time of shooting. We utilized a grand total of 46,453 street View photographs, which necessitated a substantial amount of effort to assess the weather conditions depicted in each image. To streamline this process, we conducted an analysis on a randomly selected subset of the photographs (the ones we gave our volunteers to rate). It is worth noting that the majority of these photographs were captured under clear or dry weather conditions, rather than cloudy or rainy conditions. Still, ensuring that all 46,453 photos were captured on a day with clear skies is challenging. As acknowledged by the reviewer, weather conditions do indeed impact the plant color richness (PCR). So, we have incorporated this information into the Discussion section of our paper, indicating it as one of the limitations of our study (Page 18, Lines 668-670).
- Introduction. The authors on line 76 tell the reader they will discuss two issues but the second issue is buried in the paragraph. “First” on line 78 starts a paragraph. “Second” on line 96 should start a new paragraph to help the reader.
The authors on lines 118 through 124 make statements but the article itself falls short. These sentences are overstatements. The findings also need to be clearer and more closely duplicate the work of the Kaplans (citation 44 line731-732). In the Kaplan’s many studies, they showed pictures of what was preferred. (Jumping ahead to the images Figures 6, 7, 8, 9, and 10, those figures in isolation have little meaning. It would be better to have a picture of a street in full leaf and then show the emotion related to that picture. It is understandable that the graphs are responses to multiple data sets, so perhaps several examples could be shown of what street views resulted in pleasure, relaxation, boredom, or anxiety.)
Response: We appreciate your thorough review and wise recommendations. Your careful examination and constructive suggestions have greatly contributed to the refinement of our work. We have each individually made the necessary adjustments, which are shown below.
- We have taken the reviewer's advice into consideration and rectified the problem of the second issue being obscured within the paragraph. To enhance readability, we have delineated two distinct issues in two individual paragraphs (Page 3, Lines 9).
- In accordance with the reviewer's recommendation, we have examined the claims in lines 118 to 124 of the original article. The claim that "the research investigated the mechanism by which street greening visually influences emotional perception" is clearly exaggerated. As noted by the reviewers, the examination of 'impact mechanisms' is an intricate endeavor that encompasses multiple areas of study. Our study, however, is inadequate for effectively exploring this process. We have modified the text to write "the research explores the non-linear correlation between various indicators of greening and emotional perception" (Page 3, Lines 124-125).
- According to the helpful suggestion of the reviewer, we have incorporated street view images depicting various levels of greening conditions together with their related emotional perception scores into the Discussion section. This addition aims to enhance readers' comprehension of our research (Page 15, Lines 535-540; Page 16, Lines 541-543; Page 16, Lines 568-570; Page 17, Lines 588-596).
- Methods. On lines 169-171, the authors discuss the four images used to create a panoramic picture. All of the pictures show trees in full leaf. Somewhere in the text the authors need to indicate that they chose images during specific months. Also, it is unusual to have panoramic images because a person can only look forward. It is possible on a street to have a positive view in one direction and a negative view in the opposite direction. It is rare to have a panoramic view that is similar all the way around. The authors need to discuss why they chose to use panoramic images because people only have eyes on one side of their head.
Response: Thank you for your helpful suggestions. We carefully edited our work to increase the manuscript's accuracy. The precise recommendations and their replies are mentioned below.
- In response to the reviewer's recommendation, we have included the specific period at which the street view images were shot and clarified that we have removed or substituted the winter photos (Page 4, Lines 169-172).
- We appreciate your meticulous evaluation of our work and the insightful feedback you have provided. In response to your inquiry regarding the utilization of panoramic images, we would like to provide a more detailed explanation of our decision. We selected panoramic photos for our research based on two primary rationales: First, panoramic images have the capability to offer a comprehensive and encompassing visual experience. Through the use of panoramic images, we can effectively collect and evaluate the entirety of the surrounding environment, encompassing all directions that may potentially influence human emotions. Although the human eye is oriented towards the front, our perception of the surrounding world is not solely limited to what we see in front of us. The collective visual and environmental stimuli compose the perceptual experience of individuals about the world around them, as processed by their perceptual system. Panoramic images enhance the simulation of people's perception in the real environment by offering comprehensive visual information from all directions. Second, the utilization of panoramic images is crucial for maintaining experimental control. In the real street setting, various directions may offer distinct views. By opting for panoramic images, we can ensure the uniformity of the study setting. By employing this approach, we can more effectively assess the influence of street greening without mitigating the influence of varying landscape alterations from different orientations. Ensuring the validity of the experiment is crucial.
In accordance with the reviewer's suggestion, we have provided a concise rationale for the utilization of panoramic photos and incorporated it into the Method section (Page 5, Lines 179-183).
- Discussion. This section would be helped by having pictures of what a viewer might have seen and the related emotional response. On lines 487-491, the authors discuss having too much greenness and reducing the penetration of light. In the images the authors showed, sky was always visible so there would not be a perception of oppressiveness because the viewer could see far.
Response: We appreciate your evaluation and the valuable recommendations you have provided. According to the suggestion, we added street view images into the Discussion section to enhance readers' comprehension of the relationship between participants' emotional experience and the visual stimuli they were exposed to (Page 15, Lines 535-540; Page 16, Lines 541-543; Page 16, Lines 568-570; Page 17, Lines 588-596).
- Conclusion. The authors could look at the points written for the Abstract and provide more information. The authors might offer guidance about what to have in the off season for plants when the streets cannot be fully green and in full leaf. Resident needs to have emotional well-being 12 months out of the year and not just when it is warm and sunny. While this could be in the Limitations section, the reader would be more appreciative if the authors acknowledge the need to help residents all year. There are findings for this in the data.
Response: Thank you for your careful review and insightful suggestions. Based on the reviewer's suggestions for the Conclusion and Abstract chapters, we have implemented the following modifications: 1) We have provided more clarification about the specific time period that is the focus of our street greening research (Page 1, Lines 17-18; Page 25, Lines 692). 2) In section 4.2.2, we have added the planting guidelines for cold weather. Considering the positive emotional effect of street greening, we suggest using evergreen plants like Camphor, Osmanthus fragrans, and pine to provide a consistent and pleasant emotional experience for people all year round. This advice aims to highlight that even in the winter season, when plants are not actively growing, certain evergreen species may nevertheless offer visual greenery and enduring mental benefits to inhabitants. In addition, we provide a concise summary of this information in the Conclusion section (Page 18, Lines 654-659; Page 25, Lines 711-713).
Finally, we sincerely thank the reviewer for such detailed and helpful comments and suggestions, which have made the paper more rigorous and academic, and we hope that our revision above will meet the reviewers' expectations.
Reviewer 4 Report
Comments and Suggestions for Authors
General comment
The reviewed paper touched upon the field of landscape architecture and environmental psychology. It lies well within the scope of the journal because how urban dwellers perceive the environment is clearly related to urban forestry.
Specific comments
Abstract
Line 18 to 21
It is recommended that the study objectives should be explicitly stated.
Line 25
Two SHAP values have been stated here, namely 0.225 and 0.265.
To improve the interpretation of the numbers, the authors may have to state the minimum and the maximum values, if any, of the SHAP values.
Line 26 to 27
To accurately quantify the nature and the strength of the relationship between the indicators and the perception, the authors may need to provide the relevant information, e.g. correlation coefficient.
Introduction
Line 30
There are eight keywords. The editor should make sure that the number of keywords are within the allowed number of keywords for the journal.
Line 57 to 58
How a photograph is taken has a direct impact on the landscape objects that are contained within the frame of the photograph. The authors can review methodological information related to the standard practice of using photographs in environmental psychological research.
Line 100 to 101
After reviewing the literature, the authors concluded that linear relationships may be inadequate in explaining how human emotions are related to the external environment. Yet, after the development of the SHAP indicators, not many breakthroughs have been made. Therefore, it is suggested that the authors should handle the argument in Line 100 to 101 carefully.
Methods and materials
Line 128
I have reviewed two papers which are also based on Hangzhou recently. From the research materials and photographs, it is observed that Hangzhou is a rather diverse city in terms of its landscape.
Therefore, the authors of this manuscript are suggested to ensure that the selection of the sample points for the purpose of this study is representative and objective.
Line 180
The potential application of DeepLabv3+ should be reviewed before its use in this study.
Line 185
Urban landscape is a socio-cultural phenomenon. The history, population, and planning policies of a city are modifying forces of a city’s landscape. Therefore, the suitability of the use of the ADE20K data should be justified in the case of this study.
Results
Line 369
The text in each sub-figure is too small to be legible. Please revise.
Line 403
Have the authors mistakenly swapped the x-axis label with the y-axis label in each sub-graph?
If so, from Fig. 8d, it is clear that GVI had little effect on anxiety. But a positive relationship seems to appear when a certain threshold is passed.
Line 425 to 440
For brevity, if the effect of PCR on SHAP values is limited, the text presenting such results can be trimmed.
Discussion
Line 481 to 504
Since vegetated landscape is highly diverse, it is suggested that the authors can make use of some photographs taken during the field work as illustration in this sub-section.
Line 505 to 518
Please cite the SVF values and SHAP values found in this study as evidence of the discussion.
Comments on the Quality of English Language
Only minor editing and some polishing are needed
Author Response
Dear Reviewer,
Thank you very much for your comments and recommendations. We have carefully evaluated and included your suggestions in the paper. Kindly see the included paper for comprehensive modifications implemented in response to your feedback.
The reviewed paper touched upon the field of landscape architecture and environmental psychology. It lies well within the scope of the journal because how urban dwellers perceive the environment is clearly related to urban forestry.
Response: Thank you for your constructive comments. Based on your suggestions, we have thoroughly reviewed the contents of the manuscript to enhance the persuasiveness of the study. The precise alterations are as outlined below:
Abstract
- Line 18 to 21
It is recommended that the study objectives should be explicitly stated.
Response: Thanks to the reviewer for the careful review and helpful suggestion. We clearly defined the research aims in the abstract, as suggested by the reviewer. (Page 1, Lines 14-17).
- Line 25
Two SHAP values have been stated here, namely 0.225 and 0.265.
To improve the interpretation of the numbers, the authors may have to state the minimum and the maximum values, if any, of the SHAP values.
Response: Thanks for the reviewer’s insightful suggestion. Considering the feedback from this reviewer as well as reviewer 2 and reviewer 3, we have decided against specifying the exact value of the findings and instead opted to describe them in a general way. We have removed the mention of SHAP values of 0.225 and 0.265 from the abstract session. This decision was made because these numbers may not hold significant meaning for individuals unfamiliar with the research methodology (Page 1, Lines 21-23).
- To accurately quantify the nature and the strength of the relationship between the indicators and the perception, the authors may need to provide the relevant information, e.g. correlation coefficient.
Response: We appreciate the reviewer for conducting a meticulous review and providing valuable ideas. We acknowledge your proposal about the correlation coefficient and respect your concern. We would like to offer more explanation on the problems you have addressed. Our findings indicate that the association between green indicators and emotional perception is not a straight line, but rather exhibits a non-linear pattern. However, the correlation coefficient, which is used to quantify the strength of a linear relationship, may not be sensitive to accurately capture the non-linear relationship. Hence, in light of the reviewer's recommendation, we choose to utilize the mean SHAP value as a measure of the association between characteristics. The average SHAP value is calculated by taking the mean of the SHAP value for each feature over the whole dataset for every output (predicted value) of the model. To be more specific, the average SHAP value of a feature indicates the magnitude of the feature's average influence on the model's output throughout the whole dataset. A high average SHAP value suggests that the feature has a significant impact on the overall model output.
In response to the reviewers' constructive feedback, we have incorporated graphs and accompanying text in the Results section to illustrate the mean SHAP and enhance readers' comprehension of the indicators' correlation and magnitude (Page 10, Lines 386; Page 11, Lines 404-409).
Introduction
- Line 30
There are eight keywords. The editor should make sure that the number of keywords are within the allowed number of keywords for the journal.
Response: Express gratitude to the reviewer for the reminder. Upon further examination, we have verified that the guidelines on the quantity of keywords allowed in this journal adhere to a range of 3 to 10. Therefore, the 8 keywords align with the specified criteria. We appreciate your meticulous recommendations once more (Page 1, Lines 28-30).
- Line 57 to 58
How a photograph is taken has a direct impact on the landscape objects that are contained within the frame of the photograph. The authors can review methodological information related to the standard practice of using photographs in environmental psychological research.
Response: Thank the reviewer for the insightful comments. In response to the reviewer's recommendation, we incorporated a description of the standardization of the photographs being utilized. We reviewed the method regarding the use of images in environmental psychology research. It stated that 1) the photographs chosen should be representative and encompass many environmental settings, in order to ensure the wide application of the study findings. 2) The selected photographs should include appropriate traits that align with the research objective. For instance, if the aim is to analyze the presence of greenery along a street, the chosen photos must have these specific characteristics. 3) To maintain experiment uniformity, it is imperative to standardize the images by ensuring their size, quality, and color pattern remain similar. For our experiment, we collected samples from every street in the research area at intervals of 200 meters. This sampling density ensures adequate coverage of all the different environments in the region. Furthermore, the characteristics of street view images align perfectly with the objectives of the research. Hence, to ensure the text's simplicity, we just included a standardized depiction of the photograph. These were included in the methods section (Page 5, Lines 176-177).
- Line 100 to 101
After reviewing the literature, the authors concluded that linear relationships may be inadequate in explaining how human emotions are related to the external environment. Yet, after the development of the SHAP indicators, not many breakthroughs have been made. Therefore, it is suggested that the authors should handle the argument in Line 100 to 101 carefully.
Response: After considering the suggestions from the reviewer, we have thoroughly examined the pertinent material and made revisions to our presentation. The SHAP method offers the benefit of providing an easily understandable explanation. Each SHAP value corresponds to the "contribution" of a certain characteristic towards a particular prediction outcome. Positive values indicate an increase in the predicted value, while negative values indicate a decrease in the projected value. In the context of nonlinear connections, the outcome can be influenced not only by the individual contribution of characteristics to the output, but also by the manner in which they interact. SHAP captures these interaction effects by systematically evaluating all conceivable combinations of characteristics. These withdrawals do not pertain to the prior examination of linear connections. We strongly assert that the nonlinear explanatory framework it offers is of great significance in understanding the connection between human emotional perception and the process of environmental greening. To enhance the rigor of the paper, we have made modifications to the pertinent text of the original publication. Specifically, we state that existing research often presume that there is a linear link between human emotion perception and the environment. Nevertheless, we contend that this assumption may be insufficient to fully elucidate the profound influence of a green environment on emotions (Page 3, Lines 103-106).
Methods and Materials
- Line 128
I have reviewed two papers which are also based on Hangzhou recently. From the research materials and photographs, it is observed that Hangzhou is a rather diverse city in terms of its landscape.
Therefore, the authors of this manuscript are suggested to ensure that the selection of the sample points for the purpose of this study is representative and objective.
Response: Express gratitude to the reviewer for the meticulous review and insightful remarks regarding our paper. We highly value the feedback from reviewers and have thoroughly thought on how to guarantee the integrity of our study design in the domain of street greening. Our study is specifically centered toward street greening. We understand the reviewers' concerns on the diversity of Hangzhou's landscape. However, it is important to note that our study specifically focused on investigating the influence of street greening on human emotional experience, rather than providing a thorough analysis of Hangzhou's whole landscape. To ensure focus and uniformity, we intentionally omitted landscapes other than street vegetation from the research area. Furthermore, the choice of sample locations in this study is determined by the need for thorough coverage of the research region. In order to assure the objectivity and representativeness of the research results, we implemented a strategy where sampling sites were placed every 200 meters along the road network. This approach ensured a uniform distribution and full coverage of the sampling stations across the study region.
- Line 180
The potential application of DeepLabv3+ should be reviewed before its use in this study.
Response: Thanks to the reviewer for the careful review. Per the reviewer's suggestion, we have included an analysis of the potential applications of the semantic segmentation technique, DeepLabV3+, that we employed in the Method section (Page 5, Lines196 -198; Page 22, Lines 829-834).
- Line 185
Urban landscape is a socio-cultural phenomenon. The history, population, and planning policies of a city are modifying forces of a city’s landscape. Therefore, the suitability of the use of the ADE20K data should be justified in the case of this study.
Response: We appreciate the meticulous evaluation. As suggested by the reviewer, we included a detailed explanation of the practicality of the ADE20K dataset. This description primarily focuses on highlighting the benefits of the ADE20K dataset and its successful implementation in various urban studies conducted in China (Page 6, Lines 200-208; Page 22, Lines 837-847).
The recently included citations are as follows:
Wang, R.; Lu, Y.; Zhang, J.; Liu, P.; Yao, Y.; Liu, Y. The relationship between visual enclosure for neighbourhood street walkability and elders’ mental health in China: Using street view images. J. Transp. Health 2019, 13, 90–102. https://doi.org/10.1016/j.jth.2019.02.009
Liu, Y.; Wang, R.; Lu, Y.; Li, Z.; Chen, H.; Cao, M.; Zhang, Y.; Song, Y. Natural outdoor environment, neighbourhood social cohesion and mental health: Using multilevel structural equation modelling, streetscape and remote-sensing metrics. Urban For. Urban Green. 2020, 48, 126576. https://doi.org/10.1016/j.ufug.2019.126576
Wang, X.; Liu, Y.; Yao, Y.; Zhou, S.; Zhu, Q.; Liu, M.; Luo, W.; Helbich, M. Associations between streetscape characteristics at Chinese adolescents’ activity places and active travel patterns on weekdays and weekends. J. Transp. Health 2023, 31, 101653. https://doi.org/10.1016/j.jth.2023.101653
Wang, R.; Browning, M.H.; Qin, X.; He, J.; Wu, W.; Yao, Y.; Liu, Y. Visible green space predicts emotion: Evidence from social media and street view data. Appl. Geogr. 2022, 148, 102803. https://doi.org/10.1016/j.apgeog.2022.102803
Results
- Line 369
The text in each sub-figure is too small to be legible. Please revise.
Response: Thanks to the reviewer for the careful review and suggestions. In response to the reviewer's recommendation, we have increased the size of the text in each subgraph to guarantee optimal legibility for readers (Page 11, Lines 410).
- Line403
Have the authors mistakenly swapped the x-axis label with the y-axis label in each sub-graph?
If so, from Fig. 8d, it is clear that GVI had little effect on anxiety. But a positive relationship seems to appear when a certain threshold is passed.
Response: We express our gratitude to the reviewer for the meticulous assessment. We sincerely apologize for any confusion that may have arisen from our graphs. Based on the reviewer's feedback, we meticulously examined all the graphs included in the Results section. Upon conducting a more thorough analysis, we have verified that the x and y axes in Figure 8 (d) were accurately represented and not interchanged. Furthermore, it is important to highlight that the graphs in the Result section are produced only by the SHAP algorithm, without any human intervention, hence minimizing the likelihood of mistakes.
However, with the assistance of this reviewer, we discovered that the original manuscript failed to include the labels for the X-axis and Y-axis in FIG. 8-11X. Additionally, the position of the title for each subgraph was in close proximity to the X-axis, leading readers to mistakenly interpret the title content as the index represented by the X-axis. This also leads to misinterpretation by reviewer. We express our profound apologies. Using Figure 9(d) as an illustration, each data point in the graph represents a sample. The horizontal position of the point indicates the GVI value of the sample, while the vertical position shows the matching SHAP value. When the ordinate, specifically the SHAP value, is below 0, it signifies that the GVI of the sample has a suppressive impact on anxiety. Conversely, when the ordinate is above 0, it signifies that the GVI of the sample has a stimulating impact on anxiety.
We apologize for the lack of clarity in the preceding description regarding the graph's purpose. In the updated text, we have enhanced the explanation of the diagram in the Result section to give a more comprehensive understanding of our findings for readers. Simultaneously, we modified the graphic's title to prevent any potential misinterpretation. We trust that these modifications align with your anticipated outcomes (Page 12, Lines 432; Page 13, Lines 462; Page 13, Lines 464; Page 14, Lines 480).
- Line 425 to 440
For brevity, if the effect of PCR on SHAP values is limited, the text presenting such results can be trimmed.
Response: Thanks to the reviewer's patient review. we have reduced the length of this part by 16% as per the recommendation (Page 14, Lines 467-479).
- Line 481 to 504
Since vegetated landscape is highly diverse, it is suggested that the authors can make use of some photographs taken during the field work as illustration in this sub-section.
Response: Thanks to the reviewer for the careful review and insightful suggestions. In response to the recommendations provided by both reviewer 4 and reviewer 2, we have incorporated some pictures into the Discussion section. The images serve to showcase the variety of landscapes and aid readers in comprehending the relationship between the participants' emotional perception and the visual stimuli they encounter (Page 15, Lines 535-540; Page 16, Lines 541-543; Page 16, Lines 568-570; Page 17, Lines 588-596).
- Line 505 to 518
Please cite the SVF values and SHAP values found in this study as evidence of the discussion.
Response: Thank the reviewer for the valuable comments. We have revised the paper in accordance with the recommendations provided by the reviewer. In order to enhance the rigor of the work, we have included the SVF values and SHAP values obtained from this study as evidence in the Discussion section (Page 16, Lines 573-575; Page 16, Lines 581; Page 17, Lines 584).
Finally, thanks again to the reviewer for the dedication and professional guidance throughout the review process. These professional suggestions are of great significance to our research. We are hopeful that this revised version meets the reviewer’s expectations
Round 2
Reviewer 3 Report
Comments and Suggestions for Authors
“Quantifying the Impact of Street Greening in Full Leaf on Emotional Perception: Guidelines for Resident Well-being”
Second Review
The article is greatly improved. These suggestions are to help the reader and better guarantee that the important findings are implemented to improve well-being worldwide.
This reviewer has indicated that the article only needs minor changes because the suggested alterations can be made easily by the authors.
Title: Words in the title should be changed. Rather than “…Impact of Street Greening in Full Leaf Seasons…” this should be “..Impact of Street Greening ‘During’ Full Leaf ‘Seasons’…” Full leaf is a time period and the time would be during the full leaf time period. The reason to add “seasons” is because you are studying the streets during spring, summer, and fall. You did not only select the middle of summer.
Abstract: The authors have added during full leaf to the title and this clarification should be in the first sentence in the abstract. Like 11 “…impact of street greening during the full leaf seasons in spring, summer, and fall is important…” The reason to include spring, summer, and fall is because the reader will understand that this emotional boost is available for multiple months and not just a period in the middle of summer. The reference to spring, summer, and fall is on line 693.
Lines 17 and 18 reads correctly with “in.” “…to get street view images when plants were in full leaf.”
Lines 23 and 24. This sentence can be eliminated. “Our study addressed the limited interpretability….” This sentence is a repeat of the sentence on lines 14-15 and more insightful information could be added to the abstract for the word count. You can either say what your study will do or what your study did but not both in the short abstract. Instead, include the more “contributory” information in your article.
Change the last sentences (lines 24-27) that are too vague, i.e., “providing local planning departments with support…enriching plant communities…increasing visible greenery.” Give specifics and not generalities. How many mature trees are on both sides of the road (line 630) in a ¼ mile are shown in your pictures that are most preferred? (you do need to count trees but you can do this on some of the streets in Hangzhou) Mention the value of green color and the specific plants (line 640). Mention flowering (line 644). Build up areas with low Green Value Index (spelling out GVI) (line 646). You also have beautiful characterizations that need defining i.e. 1. Green View Index; 2. Tree-Sky View Factor; 3. Plant Color Richness and 4. Plant Level Diversity. It is okay to say what the Hangzhou Boulevard Design Guidelines resulted in spectacular streets because other cities could adopt similar guidelines. You also need to tell the reader about your emotional perception scores (A. Pleasure; B. Relaxation; C. Boredom; and D. Anxiety). Your study affirmed the value of the Hangzhou guidelines and did so by assessing the streets based on the 4 measures (1,2,3, and 4) and the emotions (A, B, C, and D). The reader, including planning departments and citizen advocates, need to have clear directions of what to do to improve their cities and emotional well-being. Add the text about adding evergreen plants for the winter months (lines 655-660) and add the text from lines 716-717 about trimming. Your study has these specifics and these specifics need to be in your abstract.
Introduction. Include text about studying the streets during full leaf seasons of spring, summer, and fall.
Line 104 needs an “s” after presume.
Lines 126-131. You do not need to repeat “non-linear” because most readers will not understand this. Without an example, even an academic has difficulty understanding. It sort of just says, “It doesn’t really work in all instances.” This isn’t as helpful as giving clear guidance. Change your last two sentences. Rather than saying, “…it could offer insights… Instead, say that data on these four measures (spell out the above to remind the readers) would give communities detailed measures to help them improve their streets and well-being. (Pretend you are a city planner or a lay person and you want to make your street better. What specific information do you need to make the streets better? Your article has this information, but it needs to be clearer for the reader. You also aligned the pictures with emotion. Repeat and spell out the 1, 2, 3, and 4 and A, B, C, and D (see text for the abstract). In the Abstract and the article, the reader needs to be told the scale in the numbers for A. B. C. and D. Is 4 “high” for Pleasure or 4 “low” for Pleasure?
Line 165. Either remove the word “first” or add a second and third in the section. The reader is looking for a second and third if you write “first.” Unless the description is short for each, each should have a paragraph.
Lines 180-184 – Terrific text because you explained the panoramic view.
Line 2.3.3 This title needs to be “Sample Population for the Emotional Perception Survey”
Lines 214-220. You would include this paragraph under this heading because this describes your “evaluations from inhabitants…” (line219-220)
Lines 221-235 This section would be under 2.3.4 because this paragraph is about human emotions. It is not about the sample population.
Lines 236-249. This section belongs under the heading for 2.3.3 as this is about the study population. You always describe the population in one section so the reader knows who offered their perceptions. Your participants are special because they are residents of Hangzhou.
Results: The authors are very familiar with your GVI, PLD, TSVF, and PCR but it is tedious for the reader to have to look back in the article to find the definitions. When the opportunity exists, give both the long version and the letters. Therefore, in 3.2.1, (line 420) the title has Plant-Level Diversity. Add (PLD) after you have the long name. Do the same with the Figure 8, 3.2.2, 3.2.3, Figure 9, Figure 10, 3.2.4, and Figure 11.
Under 4.1.1, (lines 512-620) and as mentioned in the Abstract, the reader needs to be told of the scale for the numbers. Is 4 high for pleasure when 4 might mean low? The reader can guess the scale by comparing the two panoramic examples (excellent addition) in Figure 12 and 13 but the reader should also have text that describes the scale and not have to make the determination by comparing pictures.
Figure 12 and 13. Spell out PLD and put PLD in parentheses. The same is true of all the figures.
Line 590. Figure 16. Example of T-SVF value is 0.5-0.6 has no meaning for the reader. This can be eliminated or a longer section needs to be added to explain this number.
Line 599 – Under section 4.1.4, the authors write, “As mentioned in the previous ‘chapter’ but there was no previous chapter. They might have meant the previous section but this too is vague. The authors can offer the section number instead.
Lines 653-655. Eliminate the sentences, “In summary, the objective …Additionally…” You don’t need to include these generic sentences because your article makes specific contributions to knowledge. Your data speaks for itself.
Lines 662-690. You don’t need to write “Additionally” because the limitations section is a stand along section. You can just state that your study does possess limitations. You have a first, second, and third. Make these each paragraphs to help the reader. Remember that some readers skim and you want to inform even the reader who skims and has little time. In this section, you also need to mention trimming which you have in lines 715-717. Make sure that all the recommendations for streets that improve emotions are clearly provided in the
Abstract, the Results, and the Conclusions. We need better streets!
5. Conclusions (Line 708) T-SVF value is approximately 0.5 has little value for the reader.
Your final recommendation should be for cities to develop guidelines similar to the Hangzhou Boulevard Design Guidelines but with the new design details for street greening to address emotions. Your article has these details. You don’t need to explain “bad” streets but only explain “good” streets that give people pleasure and peace of mind.
Author Response
Dear reviewer:
We would like to express our sincere gratitude for your thorough review of our paper and the valuable feedback provided. We highly appreciate your professional insights, and we have diligently revised the manuscript in accordance with your suggestions.
Once again, thank you for your meticulous review. You can find the revised document in the attached file. We hope these modifications bring the manuscript in line with the expected standards.
Replies to the reviewer’s comments:
- Tittle. Words in the title should be changed. Rather than “…Impact of Street Greening in Full Leaf Seasons…” this should be “..Impact of Street Greening ‘During’ Full Leaf ‘Seasons’…” Full leaf is a time period and the time would be during the full leaf time period. The reason to add “seasons” is because you are studying the streets during spring, summer, and fall. You did not only select the middle of summer.
Response: We express our gratitude to the reviewer for the comprehensive review and valuable recommendations. Per the reviewer's recommendation, we modified the title to " Quantifying the Impact of Street Greening During Full Leaf Seasons on Emotional Perception: Guidelines for Resident Well-being." (Page 1, Lines 2-3).
- Abstract. The authors have added during full leaf to the title and this clarification should be in the first sentence in the abstract. Like 11 “…impact of street greening during the full leaf seasons in spring, summer, and fall is important…” The reason to include spring, summer, and fall is because the reader will understand that this emotional boost is available for multiple months and not just a period in the middle of summer. The reference to spring, summer, and fall is on line 693.
Lines 17 and 18 reads correctly with “in.” “…to get street view images when plants were in full leaf.”
Lines 23 and 24. This sentence can be eliminated. “Our study addressed the limited interpretability….” This sentence is a repeat of the sentence on lines 14-15 and more insightful information could be added to the abstract for the word count. You can either say what your study will do or what your study did but not both in the short abstract. Instead, include the more “contributory” information in your article.
Change the last sentences (lines 24-27) that are too vague, i.e., “providing local planning departments with support…enriching plant communities…increasing visible greenery.” Give specifics and not generalities. How many mature trees are on both sides of the road (line 630) in a ¼ mile are shown in your pictures that are most preferred? (you do need to count trees but you can do this on some of the streets in Hangzhou) Mention the value of green color and the specific plants (line 640). Mention flowering (line 644). Build up areas with low Green Value Index (spelling out GVI) (line 646). You also have beautiful characterizations that need defining i.e. 1. Green View Index; 2. Tree-Sky View Factor; 3. Plant Color Richness and 4. Plant Level Diversity. It is okay to say what the Hangzhou Boulevard Design Guidelines resulted in spectacular streets because other cities could adopt similar guidelines. You also need to tell the reader about your emotional perception scores (A. Pleasure; B. Relaxation; C. Boredom; and D. Anxiety). Your study affirmed the value of the Hangzhou guidelines and did so by assessing the streets based on the 4 measures (1,2,3, and 4) and the emotions (A, B, C, and D). The reader, including planning departments and citizen advocates, need to have clear directions of what to do to improve their cities and emotional well-being. Add the text about adding evergreen plants for the winter months (lines 655-660) and add the text from lines 716-717 about trimming. Your study has these specifics and these specifics need to be in your abstract.
Response: We greatly appreciate the reviewer’s helpful feedback. In response to the reviewer's recommendations, we thoroughly examined and modified the abstract section to enhance its conciseness and rigor. The precise alterations are as follows:
1) In response to the reviewer's constructive feedback, we have included details during the "full leaf season in spring, summer, and fall" into the first sentence of the Abstract section. This addition aims to enhance readers' comprehension by conveying that this emotional uplift may last for many months (Page 1, Lines 12-13).
2) Following the reviewer's helpful comments, we have revised the text as follows: " we used the Baidu Map API to get street view images when plants were in full leaf." (Page 1, Line 19).
3) In response to useful suggestions from reviewers, we have removed the statement " Our study addressed the limited interpretability of current perception models and measured the emotional effects of different greening indicators " in order to make the Abstract section more concise (Page 1, Line 25).
4) Based on helpful suggestions from the reviewer, we have included detailed accounts of the quantity of mature trees present in the most frequented street view images. Since the number of trees is directly associated with aspects such as crown breadth and variety, we have decided to integrate the description of the Green View Index and the range of tree numbers to provide readers with a more specific understanding (Page 1, Lines 25-27).
5) Thank the reviewer for the helpful suggestions. As suggested by the reviewer, we demonstrate the value of green by describing the enhancing impact of a high GVI on positive emotions (Page 1, Line 25). And specific plants (Page 1, Line 30) as well as flowering plants are indicated in the guidelines section of the Abstract section (Page 1, Line 30). We provide planting recommendations for areas with low GVI as well (Page 1, Line 29).
6) Give thanks to the reviewer for the valuable recommendations. In response to the reviewer's suggestion, we included clarifications for four greening indicators (Page 1, Lines 20-22) and four emotional perceptions (Page 1, Lines 22-23).
7) In response to valuable suggestions provided by the reviewer, we have incorporated recommendations regarding the planting of evergreen plants (Page 1, Lines 34-35) and the proper trimming of vegetation in particular regions to enhance visibility (Page 1, Line 34).
- Introduction. Include text about studying the streets during full leaf seasons of spring, summer, and fall.
Line 104 needs an “s” after presume.
Lines 126-131. You do not need to repeat “non-linear” because most readers will not understand this. Without an example, even an academic has difficulty understanding. It sort of just says, “It doesn’t really work in all instances.” This isn’t as helpful as giving clear guidance. Change your last two sentences. Rather than saying, “…it could offer insights… Instead, say that data on these four measures (spell out the above to remind the readers) would give communities detailed measures to help them improve their streets and well-being. (Pretend you are a city planner or a lay person and you want to make your street better. What specific information do you need to make the streets better? Your article has this information, but it needs to be clearer for the reader. You also aligned the pictures with emotion. Repeat and spell out the 1, 2, 3, and 4 and A, B, C, and D (see text for the abstract). In the Abstract and the article, the reader needs to be told the scale in the numbers for A. B. C. and D. Is 4 “high” for Pleasure or 4 “low” for Pleasure?
Line 165. Either remove the word “first” or add a second and third in the section. The reader is looking for a second and third if you write “first.” Unless the description is short for each, each should have a paragraph.
Lines 180-184 – Terrific text because you explained the panoramic view.
Line 2.3.3 This title needs to be “Sample Population for the Emotional Perception Survey”
Lines 214-220. You would include this paragraph under this heading because this describes your “evaluations from inhabitants…” (line219-220)
Lines 221-235 This section would be under 2.3.4 because this paragraph is about human emotions. It is not about the sample population.
Lines 236-249. This section belongs under the heading for 2.3.3 as this is about the study population. You always describe the population in one section so the reader knows who offered their perceptions. Your participants are special because they are residents of Hangzhou.
Response: Thank you for your meticulous review and insightful suggestions. After thorough consideration of your remarks, we have carefully implemented the required modifications individually, which are detailed below.
1) In response to valuable recommendations from the reviewer, we have incorporated details regarding " during full leaf seasons of spring, summer, and fall "into the appropriate section of the Introduction section (Page 1, Line 42; Page 3, Lines 125-126).
2) We express our gratitude to the reviewer for the meticulous suggestions, and we sincerely apologize for our negligent writing. Following the reviewer's suggestion, we have appended the letter 's' to the word 'presume' (Page 3, Line 113).
3) Thanks to the reviewer for the careful review and helpful suggestions. We followed the reviewer's recommendation and eliminated the redundant mention of the "non-linear relationship". Furthermore, we have modified the last two sentences to explicitly convey the tangible assistance offered by the research data, furnishing comprehensive details to enhance the streets and augment the welfare of inhabitants (Page 3, Lines 138-144).
4) Thanks to the reviewer for the patient review and helpful suggestions. We followed the reviewer's advice and reiterated the spelling: 1. Green View Index; 2. Tree-Sky View Factor; 3. Plant Color Richness; and 4. Plant Level Diversity. A. Pleasure; B. Relaxation; C. Boredom; and D. Anxiety. Furthermore, we outlined the numerical scales for A, B, C, and D in the Abstract and Introduction section of the text. We also provided a comprehensive explanation of the emotional connotations associated with these values, thereby enhancing comprehension of the findings (Page 3, Lines 131-132; Page 3, Lines 139-140; Page 3, Lines 132-134; Page 1, Lines 24-25).
5) Express gratitude to the reviewer for the comprehensive evaluation. In accordance with the reviewers' suggestion, we have eliminated the term "first" in order to prevent any potential confusion (Page 3, Line 113).
6) Thank the reviewer for the valuable recommendations. As per your suggestion, we have adjusted the title to accurately reflect the content: "Sample Population for the Emotional Perception Survey." (Page 6, Line 226).
7) Appreciate the reviewers' meticulous examination. According to the reviewer's suggestion, we have repositioned the corresponding paragraph beneath the heading "Perceptions Evaluations from Inhabitants" (Pages 6-7, Lines 241-248).
8) Thanks to the reviewer for the thorough and considerate evaluation, as well as their insightful recommendations. In accordance with the reviewer's recommendation, we relocated the relevant paragraph to a position below the subsection on human emotions in order to accurately categorize the content pertaining to human emotions (Page 7, Lines 250-264).
9) Thank the reviewer for the careful review. In response to the reviewers' suggestion, we moved the relevant part to the correct subsection "2.3.3" in order to provide a more accurate depiction of the study population (Page 6, Lines 227-240).
- Results. The authors are very familiar with your GVI, PLD, TSVF, and PCR but it is tedious for the reader to have to look back in the article to find the definitions. When the opportunity exists, give both the long version and the letters. Therefore, in 3.2.1, (line 420) the title has Plant-Level Diversity. Add (PLD) after you have the long name. Do the same with the Figure 8, 3.2.2, 3.2.3, Figure 9, Figure 10, 3.2.4, and Figure 11.
Under 4.1.1, (lines 512-620) and as mentioned in the Abstract, the reader needs to be told of the scale for the numbers. Is 4 high for pleasure when 4 might mean low? The reader can guess the scale by comparing the two panoramic examples (excellent addition) in Figure 12 and 13 but the reader should also have text that describes the scale and not have to make the determination by comparing pictures.
Figure 12 and 13. Spell out PLD and put PLD in parentheses. The same is true of all the figures.
Line 590. Figure 16. Example of T-SVF value is 0.5-0.6 has no meaning for the reader. This can be eliminated or a longer section needs to be added to explain this number.
Line 599 – Under section 4.1.4, the authors write, “As mentioned in the previous ‘chapter’ but there was no previous chapter. They might have meant the previous section but this too is vague. The authors can offer the section number instead.
Lines 653-655. Eliminate the sentences, “In summary, the objective …Additionally…” You don’t need to include these generic sentences because your article makes specific contributions to knowledge. Your data speaks for itself.
Lines 662-690. You don’t need to write “Additionally” because the limitations section is a stand along section. You can just state that your study does possess limitations. You have a first, second, and third. Make these each paragraphs to help the reader. Remember that some readers skim and you want to inform even the reader who skims and has little time. In this section, you also need to mention trimming which you have in lines 715-717. Make sure that all the recommendations for streets that improve emotions are clearly provided in the Abstract, the Results, and the Conclusions. We need better streets!
Response: Thank the reviewer for the thorough review and valuable feedback on our paper. We highly appreciate the suggestions and have made the necessary revisions to enhance the readability and clarity of the article. Below are our responses to your comments:
1) Thank the reviewer for the meticulous evaluation. In response to the reviewers' suggestions, we have incorporated long names and abbreviations into the appropriate headings (Page 12, Line 434; Page 12, Line 447; Page 12, Line 462; Page 13, Line 476; Page 13, Line 478; Page 14, Line 493; Page 14, Line 496; Page 15, Line 499; Page 15, Line 529; Page 16, Line 555; Page 16, Line 558; Page 16, Line 563; Page 17, Line 588; Page 17, Line 590; Page 17, Line 608; Page 18, Line 614; Page 18, Line 618; Page 18, Line 621).
2) Thanks to the reviewer for the helpful suggestions. According to the reviewer's suggestion, we have incorporated the description of the digital scale into both the Abstract and Introduction sections. This enhancement aims to enhance readers' comprehension. In addition, we have a description of fractional scale in the Method section to ensure that the reader understands the numerical scale without having to compare the images (Page 1, Lines 24-25; Page 3, Lines 132-134; Page 6, Lines 238-240).
3) Appreciate the reviewers' meticulous review. In response to the reviewer's recommendation, we have spelled out the PLD and added the PLD in parentheses for figures 12 and 13 (Page 16, Line 555; Page 16, Line 558).
4) We appreciate the perceptive recommendations that the reviewer have provided. As suggested by the reviewer, we added an explanation to the number range of 0.5-0.6 (Page 18, Lines 608-610).
5) Thank the reviewer for the comprehensive evaluation. In accordance with the reviewer's suggestion, we have replaced "previous chapter" in the original manuscript with "Chapter 3.2.4" (Page 18, Line 622).
6) We express our gratitude to the reviewer for the thorough and detailed review. Following the reviewer's recommendation, we have eliminated the sentences "In summary, the objective... Additionally..." in order to ensure the article's conciseness (Page 19, Line 675).
7) Thank the reviewer for the careful review. According to the reviewer's suggestion, we have removed the word "Additionally" in order to ensure the clear structure of paragraphs (Page 19, Line 682).
8) Thank the reviewer for the thorough review and valuable feedback. We appreciate your attention to detail. Regarding your concern about the mention of trimming, we would like to clarify that we have indeed included this recommendation in Section 4.2 of the paper. Specifically, we stated, "In areas where the GVI is extremely high, appropriate pruning of existing plants is advisable to increase general visibility and sky visibility to reduce residents' sense of insecurity." Furthermore, we have ensured that all recommendations for streets that enhance emotions are clearly presented in the Abstract, Results, and Conclusions sections. We are committed to addressing any oversights and will carefully review the entire paper to guarantee comprehensive coverage of all suggestions (Page 19, Lines 671-673).
- Conclusion. (Line 708) T-SVF value is approximately 0.5 has little value for the reader.
Your final recommendation should be for cities to develop guidelines similar to the Hangzhou Boulevard Design Guidelines but with the new design details for street greening to address emotions. Your article has these details. You don’t need to explain “bad” streets but only explain “good” streets that give people pleasure and peace of mind.
Response: Thank you for your comprehensive evaluation of our work and the valuable recommendations offered. According to the comments, we have carefully revised our paper to improve the precision and clarity of the manuscript. The general comments and specific suggestions and their corresponding responses are listed below, and we sincerely hope that the revised manuscript will fulfill the high standards for publication.
1) Thank the reviewer for the thorough review. Regarding the comment about the mention of T-SVF values around 0.5, we have considered the reviewer’s suggestion and decided to remove this reference from the conclusion to enhance clarity and focus. We believe this modification will contribute to a more concise and direct conclusion (Page 20, Line 728).
2) We express our gratitude to the reviewer for the comprehensive evaluation and valuable recommendations. We have revised the conclusion based on the reviewer's recommendation. We suggest that urban planners and policy makers should proactively establish street design guidelines. Specifically, we have emphasized measures focused on improving street greening by expanding plant levels and increasing the amount of visible green spaces in strategic locations. These proposals strive to establish street environments that are enjoyable and conducive to residents fostering positive feelings, alleviating negative emotions, and offering more habitable urban living spaces (Page 20, Lines 728-738).